# Discovery of a Roman Quarry for Pozzolanic aggregates in the Euganean Hills Magmatic District, Northeast Italy: A stepwise archaeometric approach

Simone Dilaria[1,2], Luigi Germinario[3], Claudio Mazzoli[3], Caterina Previato[1], Milo K. Pilgrim[4], Josiah Olah[1], Jacopo Nava[3], Jacopo Bonetto[1], Michele Secco[1,2]*

1 Department of Cultural Heritage (DBC), University of Padova, Padova, Italy, 2 Inter-Departmental Research Centre for the Study of Cement Materials and Hydraulic Binders (CIRCe), University of Padova, Padova, Italy, 3 Department of Geosciences, University of Padova, Padova, Italy, 4 Department of Art and Art History, The University of Texas at Austin, Austin, United States of America

* michele.secco@unipd.it

## Abstract

This study investigates the provenance of volcanic aggregates used in Roman lime-based mortars from the theatre–bath complex at Via Scavi in Montegrotto Terme (ancient *Fons Aponi*, northeastern Italy), dated to the Early Imperial period. An integrated stepwise archaeometric approach combining petrographic observations via Transmitted Polarized Light-Optical Microscopy (TPL-OM), mineralogical analyses via Quantitative Phase Analysis-X-Ray Powder Diffraction (QPA-XRPD), bulk geochemical data via X-Ray Fluorescence (XRF), microchemical analyses via Scanning Electron Microscopy coupled with Energy-Dispersive Spectroscopy (SEM-EDS), and µ-Raman Spectroscopy were applied to characterize the volcanic components of the mortars and assess their hydraulic behaviour. The results show that the mortars incorporate angular trachytic to trachyandesitic volcanic breccias displaying well-developed reaction rims and extensive pozzolanic reactivity, leading to the formation of calcium–aluminosilicate hydrate phases typical of pozzolanic lime mortars. Comparison with a comprehensive reference database of volcanic rocks from the Euganean Hills Magmatic District (Veneto, Italy) indicates that these aggregates are consistent with explosive diatreme breccias exposed in the eastern sector of the district, most likely corresponding to quarry sites near Villa Draghi (Montegrotto Terme). The identification of these volcanic materials in mortars from Aquileia (northeastern Italy, approximately 150 km north-east of the Euganean Hills) further suggests that such "Euganean pozzolans" were not used exclusively at a local scale but were traded over longer distances. These findings provide new archaeometric evidence for the exploitation of non-Vitruvian volcanic pozzolans in Roman construction and illustrate the potential of integrated petrographic and geochemical approaches for provenance studies of these mortar aggregates.

**Data availability statement:** All relevant data are within the manuscript and its Supporting Information files.

**Funding:** MS was supported with the contribution of Fondazione Cassa di Risparmio di Padova e Rovigo as part of the Bando Ricerca Scientifica di Eccellenza 2023 [grant number 68051]. Further support was obtained in the framework of "The Geosciences for Sustainable Development" project (Budget Ministero dell'Università e della Ricerca–Dipartimenti di Eccellenza 2023–2027 C93C23002690001). The research infrastructures employed in this project were implemented and funded by the University of Padova within the World Class Research Infrastructures (WCRI) programme –SYCURI (Synergic Strategies for Cultural Heritage at Risk). The funders had no role in study design, data collection and analysis, decision to publish, or preparation of the manuscript.

**Competing interests:** The authors have declared that no competing interests exist.

## "Vitruvian" and "alternative" volcanic pozzolans in the archaeological record: state of the art

Studies on ancient construction materials are increasingly gaining prominence within archaeological research, driven by the growing integration of scientific approaches. Among these, volcanic pozzolans have attracted particular attention, as their interaction with lime-based binders played a key role in enhancing the mechanical performance and hydraulic properties of ancient mortars and concretes through the development of hydrated-calcium-silicates, hydrated-calcium-aluminates, and hydrated-calcium-aluminosilicates [1–4]. Roman treatises, including Vitruvius and Pliny the Elder, distinguish between two principal types of volcanic pozzolans used in Roman construction [5,6]. The first, *harenae fossiciae* (Vitr. 2.5.1; Plin. 36.174–175), are associated with the Pliocene cinerites of the Colli Albani volcanic district [7,8] and were extensively employed in mortars throughout Rome and central Latium from the second half of the 2nd c. BCE onward [9], with a main exploitation of the *pozzolane rosse* facies particularly during the Imperial Era [10–13]. The second type, *pulvis puteolanus* (Vitr. 2.6.1; 5.12.2; Plin. 35.166), primarily derives from pumice and non-lithified cineritic deposits of the Neapolitan Yellow Tuff and post-Neapolitan Yellow Tuff formations of the volcanic districts around the Gulf of Pozzuoli (Phlegraean Fields). Recent multidisciplinary research has confirmed the extensive use of Phlegraean pozzolans not only locally [14,15] but also within the wider Mediterranean, becoming a cornerstone of Roman concrete technology extensively disseminated across the Empire from the Augustan period onward. The use of these volcanic pozzolans has been scientifically documented in Imperial harbour structures [16] and, recently, in on-land buildings in coastal sites from Sardinia and Northern Italy [17–19]. Additional evidence for the potential use of pozzolanic products from the Gulf of Pozzuoli has been reported at several North African coastal sites, such as Utica and Hippo Regius [20,21]. However, these latter findings still require validation through comprehensive provenance studies based on the geochemical fingerprinting of specific trace elements. Collectively, these data point to a well-organized maritime distribution network, likely facilitated by the transport of volcanic pozzolans as ship ballast, with the Gulf of Pozzuoli serving as a strategic hub for large-scale distribution. Beyond these examples, several studies have already highlighted the use of non-Vitruvian volcanic pozzolans that possess a latent reactivity comparable to their more famous counterparts. The first evidence dates back before the Roman Age, with applications documented as early as the 4th-3rd c. BCE in the Punic settlement of Pantelleria [22–24], where scoriaceous products from local volcanic outcrops were used in waterproofing mortars. On the other hand, evidence of deliberate utilization of volcanic pozzolans in mortars from the Greek world, in the Classical and Hellenistic period, is more evanescent [6]. The use of non-Vitruvian volcanic pozzolans in mortar-based materials strongly developed during the Roman age, with several documented cases including Anatolia [25–28], Sardinia [29,30], the Rhineland (Trass) [31], and Late Antique Pantelleria (Scauri) [32]. The current state of the art suggests that these alternative volcanic pozzolans were mainly employed in structural mortars for monumental Roman architecture to ensure high mechanical performance

(Table 1). In contrast, they seem to have been less frequently used for waterproofing hydraulic infrastructures like cisterns or aqueducts. Chronologically, most attestations fall within the Imperial period (1st–3rd century CE), coinciding with the widespread adoption and diffusion of Roman concrete technology across the Empire (Table 1). From a geographical perspective, many of these resources were situated within the inland territories of the Roman Empire, often in close proximity to the specific sites or towns where the materials were utilized. Nevertheless, in most of the aforementioned cases, the identification of the volcanic pozzolans' origin remains largely hypothetical, restricted to general compatibilities with geological formations across extensive geographic regions. Provenance studies seeking to link materials directly to specific quarry sites are still rarely undertaken; such investigations can only be achieved when supported by extensive reference datasets and a precise characterization of the geological sources (e.g., [29]). Following this approach, the present study provides a well-documented case of the use of "alternative" volcanic pozzolans in antiquity. These aggregates were recognized in Roman mortars from Montegrotto Terme (Padova, Italy), the ancient *Fons Aponi*, and were confidently traced to a specific quarry within the nearby Euganean Hills, the main magmatic district of the Veneto region. This precise provenance determination offers profound insights into both the local exploitation and the potential regional circulation of these materials. It underscores the strategic resource management of Roman builders, reflecting a sophisticated mastery of the territory's geological landscape, from the precise localization of outcrops to the inherent availability and variability of its lithic resources, all of which were harnessed for their optimal application.

## The stone resources of the Euganean Hills Magmatic District and Database Implementation

The Euganean Hills Magmatic District (hereafter Euganean District) forms the youngest part of the Venetian Volcanic Province (VVP), whose activity spans from the Late Palaeocene to the Late Oligocene. In the area now occupied by the Euganean Hills, volcanism began only during the final stage of the VVP, on a region underlain by a Mesozoic-Paleogene carbonate platform deposited over nearly 120 million years. Two main volcanic events occurred: an older, Late Eocene

**Table 1. The main evidence of exploitation of "alternative" volcanic pozzolans other than *pulvis puteolanus* from the Gulf of Pozzuoli volcanic districts and *harenae fossiciae* from the Colli Albani volcanic district. *Legend: TPL-OM=Transmitted Polarized Light-Optical Microscopy; XRD=X-Ray Diffraction; SEM-EDS=Scanning Electron Microscopy coupled with Energy-Dispersive Spectroscopy; XRF=X-Ray Fluorescence; AES=Atomic Emission Spectrometry; AAS=Atomic Absorption Spectrometry.**

| Region | Site | Building | Chronology | Function | Pozzolan type | Provenance | Analytical techniques* | Reference |
|---|---|---|---|---|---|---|---|---|
| **Turkey** | *Nysa* | various | Roman Age (?) | Structural | Volcanic rocks | Western Anatolia (Dikili-Çandarlı)? | XRF; XRD; SEM-EDS | [25] |
| | *Aigai* | Theatre, Baths, Agora | | | | | | |
| | *Sagalassos* | various | Imperial Age (Augustan Age – 4th c. CE) | Structural | Tuffs, scattered lava | Western Anatolia (area of Lake Golcuk) | TPL-OM; XRD; AES/AAS; SEM-EDS | [26–28] |
| **Germany** | *Köln* | | 2nd c. CE | Structural | Tuffs | Rhineland (Trass) | Visual observation | [31] |
| **Italy (Sardinia)** | *Nora* | Theatre | 1st c. BCE | Structural | Obsidian/perlite | Sardinia, Mt. Arci (Perdas Urias) | TPL-OM; XRF; LA-ICP-MS | [29] |
| | | Western Baths | 2nd c. CE (scattered) | Structural | Obsidian | Sardinia (Mt. Arci) | TPL-OM; SEM-EDS | [30] |
| **Italy (Pantelleria)** | *Scauri* | | 4th c. CE | Plasters | Volcanic scoria | Pantelleria (Cuddia Rossa) | TPL-OM; SEM-EDS | [32] |
| | *Cossyra* | Cisterns | 4th c. BCE | Cistern coatings | Volcanic scoria | Pantelleria (Cuddia Rossa) | TPL-OM | [22–24] |
| | | | 3rd – 2nd c. BCE | Cistern coatings | Volcanic scoria | Pantelleria (Cuddia Bruciata + Cuddia Rossa) | TPL-OM | |

phase (42 ± 1.5 Ma) characterized by submarine mafic volcanism with pillow lavas, breccias, and hyaloclastites, and a younger Oligocene phase (34–32 Ma) marked by the emplacement of more evolved magmas, primarily trachytes and rhyolites, with minor amounts of latites, in domes, plugs, dykes, and laccoliths [33–35]. This latter stage was also accompanied by submarine phreatomagmatic explosions, which produced pipe-like diatremes subsequently infilled with volcanic breccias. These deposits contain angular clasts and juvenile fragments in a tuffaceous matrix [36], primarily outcropping in the central-eastern part of the Euganean District.

The diverse eruptive activities of the Euganean Hills have profoundly shaped the flatlands of the Po Plain and have been exploited as valuable stone resources since pre-Roman times [37 and references therein]. These volcanic products were used not only as raw material for construction and artifact production but also, in some cases, as ceramic temper [38,39]. Among them, the Euganean trachyte emerged as a particularly appreciated variety [40]. Extensively quarried from antiquity until the mid-20th century CE and with only few extraction sites still active today [41], it became one of the most important stone resources of the region. During the Roman period, trachyte exploitation intensified considerably. Numerous quarry sites were opened and cultivated for centuries as the stone was widely used in road paving, hydraulic infrastructure, building walls, funerary monuments, and architectural elements in Roman cities of Northern Italy [40 and references therein]. Previous analytical studies of artifacts made from Euganean trachyte demonstrated that the main quarrying areas can be distinguished by their petromineralogical and geochemical signatures, allowing researchers to link artifacts to major extraction sites such as Monselice, Monte Merlo, and Monte Oliveto [e.g., 42, 43, 44, 45]. Building on these foundations, over the last decade the University of Padova has developed a high-resolution reference database of Euganean quarry sources through systematic sampling of both active and historic quarry fronts [41]. Geological samples collected from these sites have been incorporated into a comprehensive reference database, currently comprising around 280 samples that encompass the full range of Euganean lithologies exploited in the past. Samples were georeferenced into a tailored GIS database, developed in QGIS environment (Fig 1, a), and organized according to lithological categories which were fully characterized via TPL-OM, QPA-XRPD, and XRF analyses. In addition to the major quarries, the dataset now also includes samples from numerous smaller and long-abandoned extraction sites, which are often absent from historical maps and archival sources [e.g., 47]. These sites, mainly located in the eastern and southeastern Euganean Hills, have recently been identified through UAV-LiDAR mapping (Fig 1, b–f). Analysis of the resulting digital terrain models (DTMs) revealed slope anomalies, convexities, and concavities inconsistent with natural landforms, ultimately leading to the identification of approximately 90 previously unrecorded quarry sites, where, in most cases, traces of pre-industrial extraction toolmarks were identified after on-site field surveys [48]. This database now provides a robust framework for tracing volcanic stones in ancient artifacts and architectural remains throughout the Veneto region and neighbouring areas. Recent research has further demonstrated that, alongside trachytes, rhyolites were also exploited by Roman and pre-Roman builders [49]. These findings offer new insights into the intensity of ancient quarrying, the criteria guiding material selection, and the intricate socio-economic patterns underlying local and regional trade networks involved in the circulation of these stone building materials [41,49–52].

## The Roman complex in Via Scavi (Montegrotto Terme)

The Via Scavi archaeological area in Montegrotto Terme, located around 2 km east of the central-eastern portion of the Euganean Hills (Fig 2, a), represents one of the main Roman sites in the Veneto region and includes a bath complex and a small theatre/*odeon* [53,54]. The site was first discovered and excavated between 1781 and 1788, with further excavations carried out in 1953, the 1960s, 1970, and 1994–1995. The bath complex comprises 3 pools (A, B, and C) surrounded by richly decorated rooms and porticoes, a separate building (D) equipped with a circular basin in the centre and several auxiliary water-supply infrastructures such as structure H, which housed a *noria* (waterwheel). In the northern sector of the site, a theatre was located (E), possibly used as an *odeon*, featuring a 28m diameter and a small temple or tribune positioned at the top of the *cavea* (seating tiers). The theatre is preserved only at the foundation level, and all

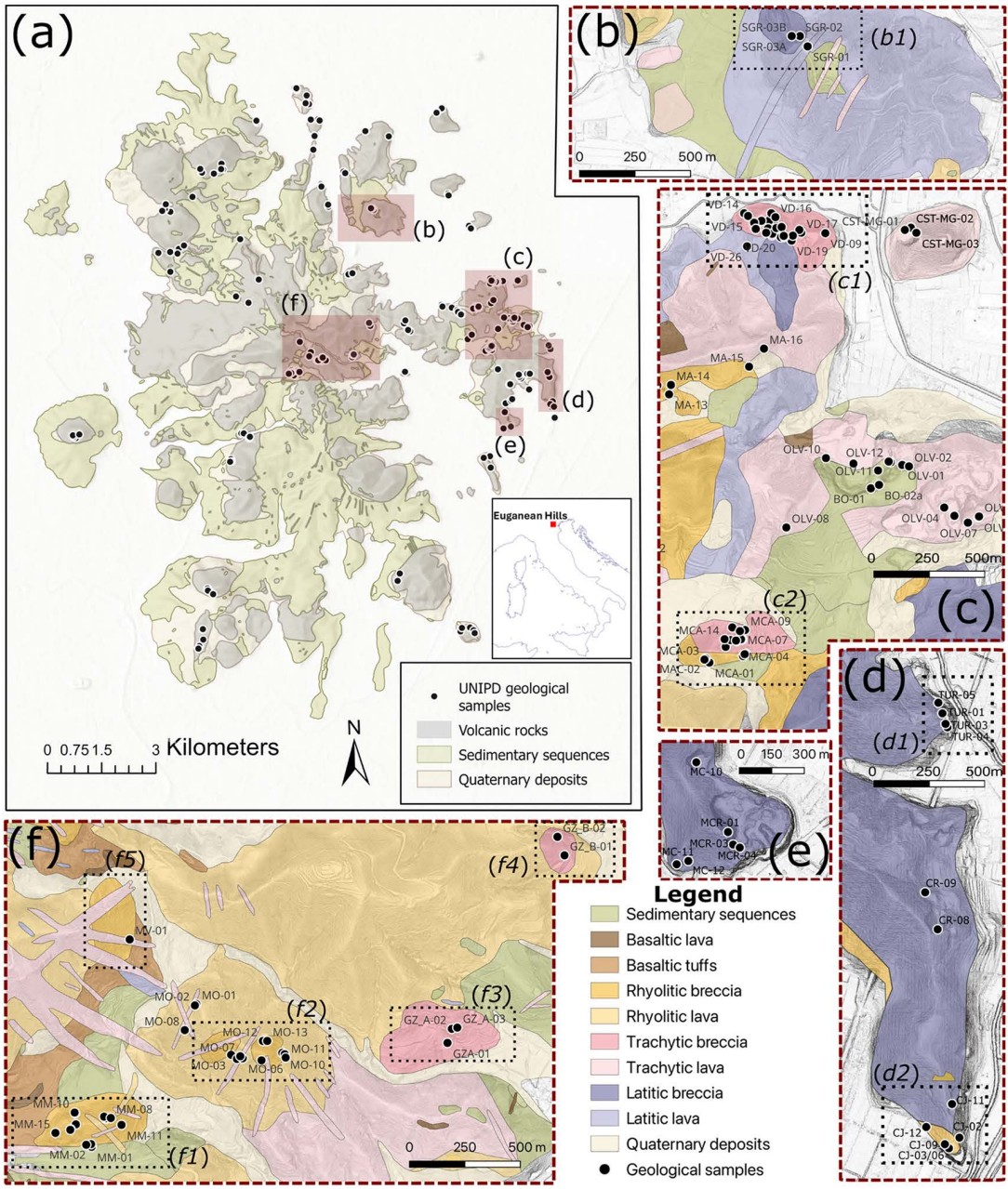

**Fig 1. (a) Simplified geological map of the Euganean Hills Magmatic District, including the positioning of the geological samples constituting the reference database developed by the University of Padova; the figure was entirely produced by the authors.** In the red-dashed boxes the localization of the main quarries documented via UAV-LIDAR and on-site surveys is reported: (b) North-eastern area; (c) Central-eastern area, northern portion; (d) Central-eastern area, southern-eastern portion; (e) Central-eastern area, southern portion; (f) Central area. Black numbers refer to the sites from which rhyolitic/trachytic/latitic explosive diatreme breccia and submarine breccia samples were collected, discussed in the text: (b1) M. Sengiari outcrop; (c1) Villa Draghi outcrop; (c2) Via Scagliara di M. Castellone outcrop. Differently from geological map [46] reporting rhyolitic composition, our investigations demonstrated that the northern portion of M. Castellone outcrop (Via Scagliara) exhibits trachytic breccia composition; (d1) Turri north of M. Nuovo outcrop and (d2) Castel del Catajo zone; (e) M. Croce outcrop; (f1) M. Marco outcrop; (f2) M. Orsara outcrop; (f3) Outcrop of Galzignano A; (f4) Outcrop of Galzignano B; (f5) M. Venda outcrop.

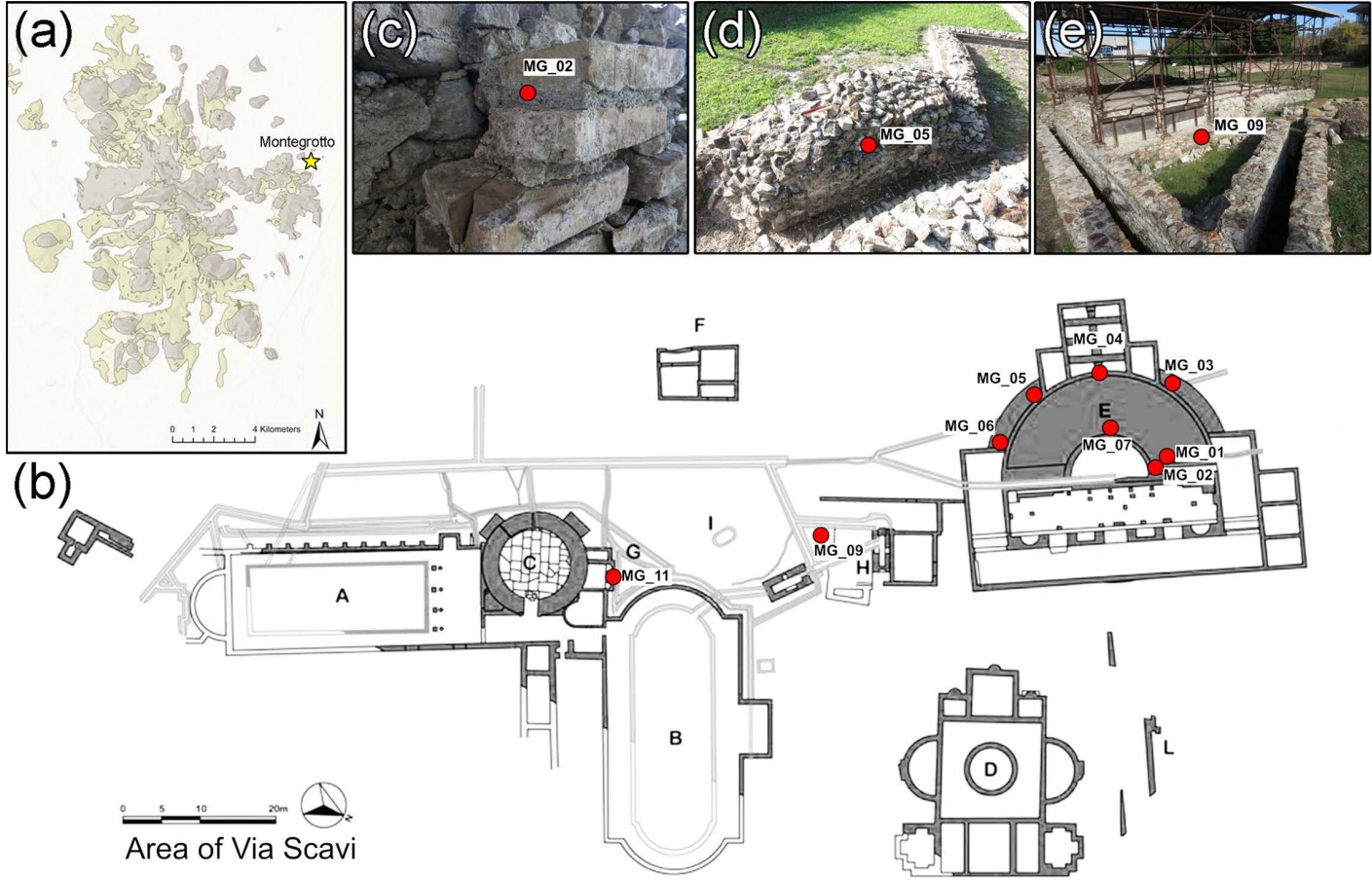

**Fig 2. Mortar samples collected from the Via Scavi archaeological area in Montegrotto and their position in the plan.** (a) Positioning of Montegrotto in respect to the Euganean District (for geological color legend see Fig 1); (b) the archaeological area of Via Scavi with positioning of the analysed mortar samples (the labelling of the structures is reported according to [53,54]). The archaeological site basemap has been edited and reprinted from [https://www.aquaepatavinae.it/portale/?page_id=1690&recid=44] under a CC BY license, with permission from © Department of Cultural Heritage of the University of Padova, original copyright [2011]; (c – e) Representative images of the analysed structures with exact position of the sampled points.

decorative elements have been removed. Both the bath complex and the theatre were built in the early Imperial age (first half of the 1st c. CE) and underwent restorations and renovations during the mid-Imperial period.

Nine pluricentimetric samples of structural mortars (Fig 2, b) originating from wall joints and *opus caementicium* structural cores were collected using hammer and scalpel (Fig 2, c-e) and analysed in the present study (Table 2). All necessary permits were obtained for the described study, which complied with all relevant regulations

A multi-analytical archaeometric approach was employed, comprising petrographic characterization of mortars and geological samples prepared on 30 μm thin sections by Transmitted Polarized-Light Optical Microscopy (TPL-OM) and microchemical analyses through Scanning Electron Microscopy coupled with Energy Dispersive Spectroscopy (SEM-EDS). Moreover, Quantitative Phases Analyses by X-Ray Powder Diffraction (QPA-XRPD), chemical analysis by X-Ray Fluorescence (XRF) on powdered samples, and μ-Raman spectroscopy were employed. This approach aimed to (1) characterize the composition and raw materials of the mortars, with particular attention to the volcanic pozzolans they contain; (2) evaluate their hydraulic properties and pozzolanic phases to reconstruct production technology; and (3) determine the

**Table 2. The analysed samples from Via Scavi site, with indication of the structure of origin and the function of the mortar (the labelling of the structures is reported according to [53,54]).**

| Sample | Building | Structural Element | Technique | Function |
|---|---|---|---|---|
| MG_01 | Theater (E) | Substructure of the *cavea* | *Opus caementicium* | Structural mortar |
| MG_02 | Theater (E) | Inner wall of the *cavea*, facing | Brick masonry | Bedding mortar |
| MG_03 | Theater (E) | External perimetral wall of the *cavea*, core | *Opus caementicium* | Structural mortar |
| MG_04 | Theater (E) | Substructure of the temple/tribune, flying buttress, core | *Opus caementicium* | Structural mortar |
| MG_05 | Theater (E) | External perimetral wall of the *cavea*, core | *Opus caementicium* | Structural mortar |
| MG_06 | Theater (E) | External perimetral wall of the *cavea*, core | *Opus caementicium* | Structural mortar |
| MG_07 | Theater (E) | Substructure of the *cavea* | *Opus caementicium* | Structural mortar |
| MG_09 | Baths, Hydraulic structure | Perimeter wall (collapsed), facing | Brick masonry | Bedding mortar |
| MG_11 | Baths – Room (4) | Perimeter wall, core | *Opus caementicium* | Structural mortar |

exact geological provenance of the pozzolans at the highest possible resolution, thereby tracing ancient quarry fronts and the distribution pathways of these materials. Details on the instrumental equipment are reported in S1 File.

## Results

### Characterization of mortars and raw materials

The overall petrographic features of the mortar samples (binder, lime lumps, aggregates) were initially described via TPL-OM. All samples resulted to be lime-based mortars incorporating abundant volcanic aggregates, macroscopically visible from the fresh cuts of the samples (Fig 3, a).

At the optical microscope, in several cases, volcanic fragments exhibit well-defined and pronounced low-birefringence reaction rims at their interfaces with the binder, indicating a high degree of reactivity (Fig 3, b). In smaller clasts (<400 μm), the pozzolanic reaction often leads to the near-complete alteration of the original material, leaving only certain microlites and phenocrysts unaltered. Only a few volcanic grains show no reaction rims and do not act as pozzolans in the samples (Fig 3, c). These volcanic aggregates share very similar mineralogical and textural characteristics, but a slight variability in some petrographic features suggests the co-existence of fragments from different lithic materials. In general, volcanic aggregates are characterized by porphyritic, or feebly glomeroporphyritic, texture with low phenocryst-groundmass ratio (Fig 3, d), constituted by feldspars (mainly plagioclase), followed by anorthoclase and sanidine, biotite, and brown amphibole (kaersutite-pargasite-sadanagaite in composition), as suggested by the SEM-EDS analyses (S2 Table); accessory phases include apatite and opaque minerals including magnetite, as indicated via SEM-EDS (S2 Table), and subordinate ilmenite. Most fragments observed in the mortars consist primarily of groundmass domains, with variability mainly in the degree of crystallinity (hypo- to holocrystalline) and the grain size of the matrix (crypto- to microcrystalline); the feldspar microlites embedded within the matrix generally display a felty texture. A secondary component of the aggregate consists of well-distributed, rounded siliceous and carbonate sand-sized grains, dominated by monocrystalline quartz, subordinate lithoclasts of quartzite, chert, and occasional gneiss, granitoids and sandstones (Fig 3, e). Isolated grains may also include mica lamellae (biotite and muscovite), chlorite, and feldspar. Carbonatic sand is less abundant, comprising micritic limestones and dolostones, often displaying reaction rims indicative of de-dolomitization [55]. The aggregates therefore display a distinctly bimodal texture. The sand fraction, originating from local alluvial deposits [56] as suggested by the rounded shape of the clasts, is medium- to fine-grained (500–125 μm; grain-size are classified according to [57]). On the other hand, the volcanic clasts are coarser (plurimillimetric) and angular, indicating mechanical fragmentation. Occasional angular ceramic fragments were also observed in some samples, such as in MG_02. Sample MG_11 stands out for its higher content of fine quartzite, chert, and dolomitic aggregates, although these are less uniformly distributed compared to the volcanic fraction. In most samples, the binder shows zoning rather than uniform birefringence, likely reflecting in

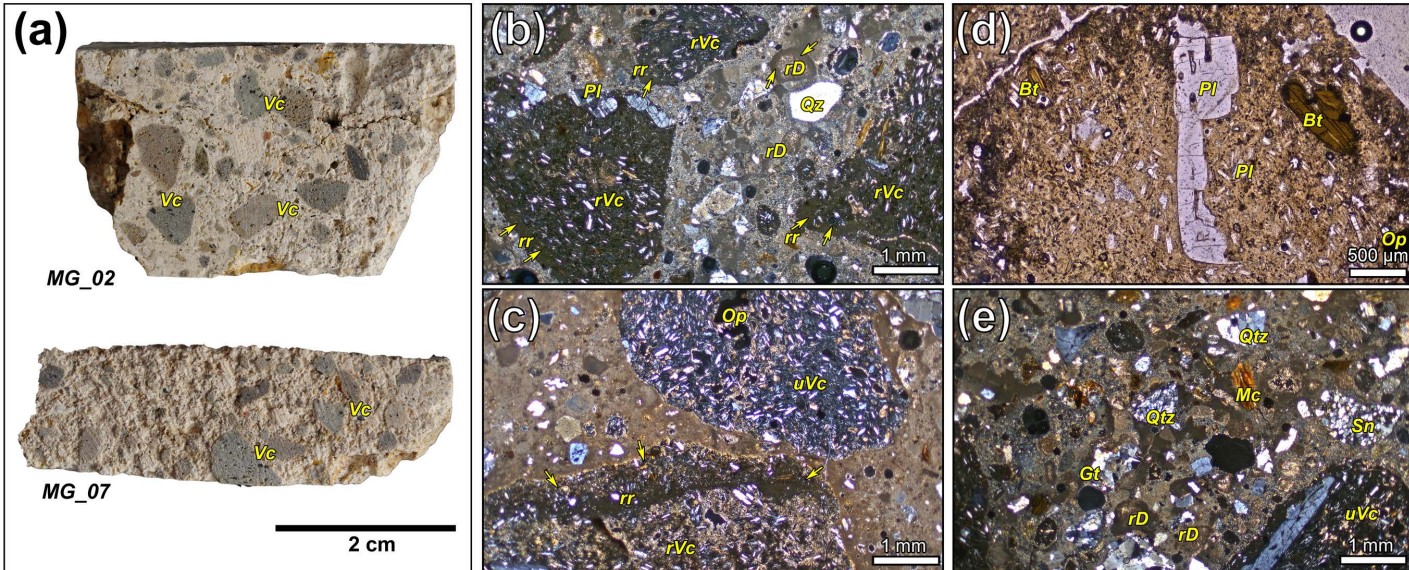

**Fig 3. Mortar samples from Via Scavi archaeological area.** (a) fresh crosscut of two representative samples; (b-e) micrographs acquired in TPL-OM (XP and PP) of representative portions of the mortar samples. Legend: Vc = volcanic clasts, rVc = reacted volcanic clast; uVc = unreacted volcanic clast; rr = reaction rim; rD = reacted dolostone; Qz = quartz; Qtz = quartzite; Gt = granitoid, Mc = mica; Sn = sandstone; Bt = biotite; Pl = plagioclase; Op = opaque mineral.

situ development of hydrated phases even within the mortar matrix. Occasional lime lumps were observed in all samples except MG_09, typically exhibiting low interference colors. No remnants of incomplete calcination were identified. The binder-to-aggregate proportion has been visually estimated at approximately 1:2; only in MG_11 it is higher, indicating a lime-rich mortar. Regarding porosity, vugs and vesicles are common throughout the samples, while planar shrinkage voids are nearly absent.

## Pozzolanic properties

The possible development of pozzolanic reactions in the mortars and the diffusion of hydraulic phases within the binders, inferred from TPL-OM observations, were further investigated through microchemical analyses using Scanning Electron Microscopy with Energy Dispersive Spectroscopy (SEM-EDS). SEM-EDS analyses of interface profiles between reactive aggregates and binder matrices confirmed that most volcanic clasts reacted with the lime binder, as shown by clear interfacial alteration zones ranging from 50 to 100 μm. These areas are characterized by significant Ca-enrichment and corresponding Si-depletion (Fig 4, a-a2), testifying the development of C-(A)-S-H products, with a progressive increase in the Ca:Si ratio towards values approaching 1:1 in the binder matrix (Fig 4, a3). The binder exhibits trends consistent with TPL-OM observations: zones displaying high interference colors correspond to areas with elevated Ca and reduced Si (Fig 4, b-c2). In contrast, the surrounding matrix shows low interference colors and consistently exhibits high Si/Ca ratios, indicating that C-(A)-S-H phases developed from the clast-binder interfaces throughout the domains of the binder matrix.

These findings confirm that the reaction of volcanic clasts was the primary driver for the formation of Ca-based pozzolanic phases. In contrast, the contribution of Mg ions, released through the dedolomitization of dolostones and dolomitic limestones, had only a minor influence on the development of magnesium silicate hydrates (M–S–H), a process observed more prominently in other contexts [4].

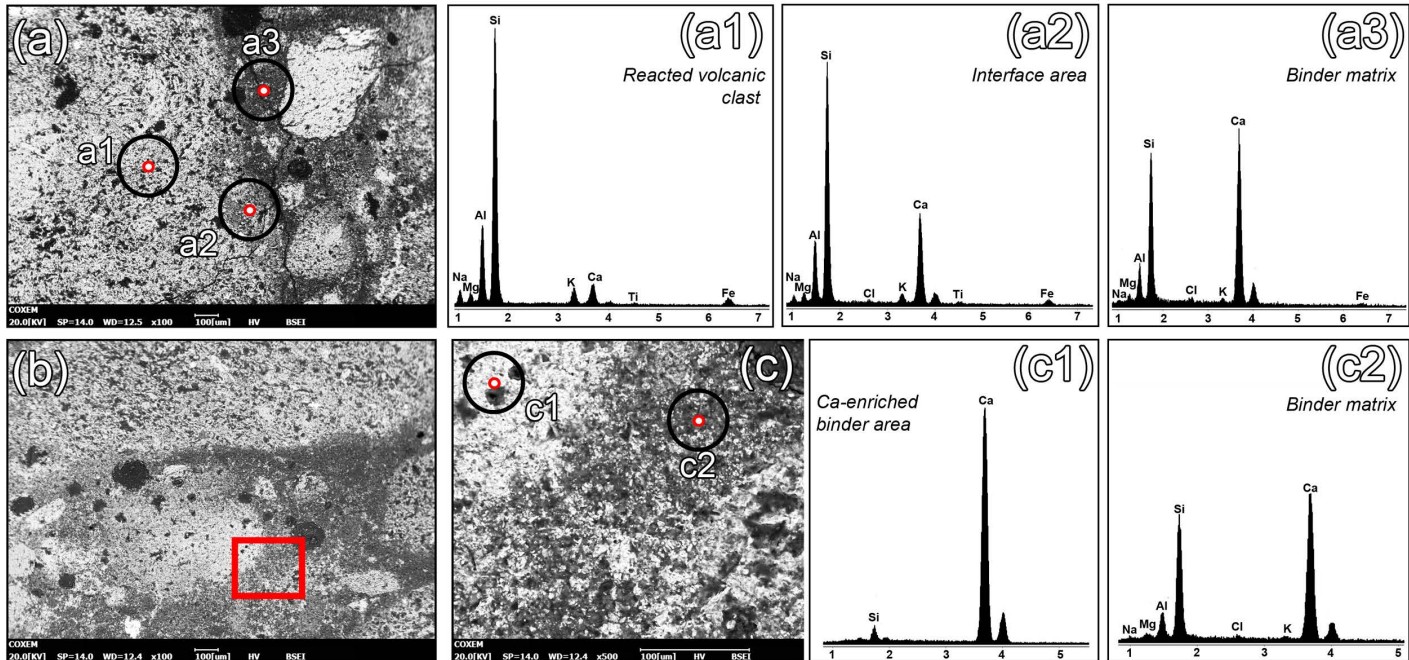

**Fig 4. SEM Backscattered Electron (BE) images and EDS microanalyses of the analysed mortars.** (a) SEM-BE acquisition of an area of sample MG_02, between a reacted volcanic aggregate and the lime binder; (a1) EDS analysis of an unreacted zone of the volcanic groundmass within a volcanic aggregate; (a2) EDS analysis of an interface area, with enrichment in Ca from the lime binder and possible local development of C-A-S-H phases; (a3) EDS analysis of the matrix of the samples, indicating pronounced Si and Al peaks suggesting the diffusion of C-A-S-H products in the binders; (b) SEM-BE image of a Ca-enriched area (lime lump) in the binder matrix of sample MG_02; (c) magnification of the area indicated by the red square in Fig (b); (c1) EDS analyses of a Ca-rich core of the lime lump; (c2) EDS analysis of the binder matrix area surrounding the lump, with clear Si and Al peaks indicating the diffusion of C-A-S-H products in the binder.

To better characterize the nature and, where applicable, the degree of crystallinity of the reaction products, Quantitative Phase Analysis by X-Ray Powder Diffraction (QPA–XRPD) was performed on separated binder fractions from two samples, MG_01-binder and MG_05-binder (Fig 5 and Table 3), following the Cryo2Sonic separation procedure adapted with a chelating agent as reported in S1 File. The analysis confirms the development of newly-formed AFm phases (Alumino-Ferrite mono-substituted) [59,60]. These crystalline phases are typical products of pozzolanic reactions between the reactive aluminosilicates of the volcanic clasts and the lime binder, specifically indicating the formation of calcium aluminate hydrates (C-A-H). Moreover, vaterite, a metastable polymorph of calcium carbonate known to form as a secondary product of pozzolanic reactions in lime-based mortars, was frequently detected in the samples. Its occurrence, linked to lime carbonation in silicate-rich systems, suggests ongoing or incomplete crystallization processes preceding the formation of stable calcite [61]. The predominance of amorphous material, together with the low abundance of crystalline calcite, indicates that most pozzolanic reaction products occur as C-(A)-S-H gel with variable stoichiometry, as typically observed via XRPD for the hydraulic phases developed after the interaction between volcanic pozzolans and Ca-based binders [4].

## Provenance determination of the volcanic clasts

The volcanic clasts in the Montegrotto mortars were traced to the Euganean Hills Magmatic District, showing petrographic, mineralogical, and geochemical features consistent with the volcanic products from this area [36,41]. Notably, the Euganean District represents the largest and most exploited local stone source and the nearest extraction site to Montegrotto [40]. However, considering the very small size of the volcanic fragments and their peculiar texture, it is not possible

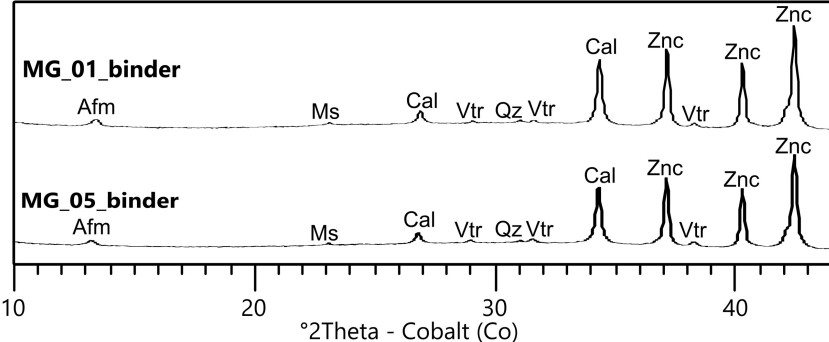

**Fig 5. XRPD spectra of the separated binder fraction of sample MG_01 and MG_05.** Mineral phases labelled according to [58] when mentioned: Ms = muscovite; AFm = AFm phases; Cal = calcite; Vtr = vaterite; Znc = zincite (internal standard).

**Table 3. Results of the QPA-XRPD analyses on binder-separated fractions of lime binder.**

| Sample | Calcite | Vaterite | AFm | Quartz | Muscovite | Amorphous |
|---|---|---|---|---|---|---|
| MG_01-binder | 23,5 | 0,8 | 3,7 | 0,4 | 0,5 | 71,1 |
| MG_05-binder | 22,6 | 1,9 | 2,4 | 0,3 | 0,5 | 72,3 |

to further refine the provenance attribution and trace a possible source quarry area based uniquely on the petrographic features of the grains.

The provenance of the volcanic pozzolans was therefore determined through a geochemical approach. Ten clasts, mechanically extracted from the mortars and polished according to the procedure described in S1 File, were analysed by X-Ray fluorescence (XRF) to obtain their bulk-rock geochemical composition (S1 Table). Previous studies have shown that bulk XRF geochemistry can provide a reliable first-order, and in some cases conclusive, tool for discriminating Euganean volcanic resources at the quarry-district scale [49–51]. This approach proved particularly effective due to the presence of a set of highly diagnostic chemical elements whose relative abundances remain coherent even in small clasts and heterogeneous fabrics. The selection of such discriminant elements was informed by previous successful provenance studies of Euganean volcanic rocks [41,49], which successfully utilized bivariate scatterplots of major and trace elements to geochemically distinguish potential source areas.

Based on the TAS (Total Alkali vs. Silica) diagram for classification of volcanic rocks [62], three clasts (MG_03-c3, MG_03-c4, MG_03-c2) fall entirely within the trachyte field (Fig 6), exhibiting a $SiO_2$ content ranging from ~ 63 to ~ 64%ox., and $Na_2O + K_2O$ around ~ 10%ox. (T-group). Most of the remaining clasts plot within the transition zone between trachytic and trachyandesitic compositions (MG_03-c5, MG_05-c1, MG_07-c1, MG_09-c1 and MG_09-c4), hereafter defined as "T-TA group." This is due to their lower silica content (ranging from ~60 to ~61%ox.) and reduced total alkalis, particularly their low $K_2O$ content (consistently below 4.0%ox.). Within the T-TA group, we also included the two remaining clasts (MG_01-c1 and MG_01-c2), plotting towards the basaltic trachyandesite field; their chemical signature remains consistent with that of volcanic clasts that have undergone reaction with the binder: their higher LOI indicates volatile release, mainly from carbonate decomposition, leading to Ca enrichment due to alteration, interaction with surrounding lime binder, and partial mobilization of Ca from residual fluids in clast micropores. QPA–XRPD of MG_01-c1 reveals high calcite content (Table 4), confirming that these chemical modifications stem from pozzolanic reactions.

                    

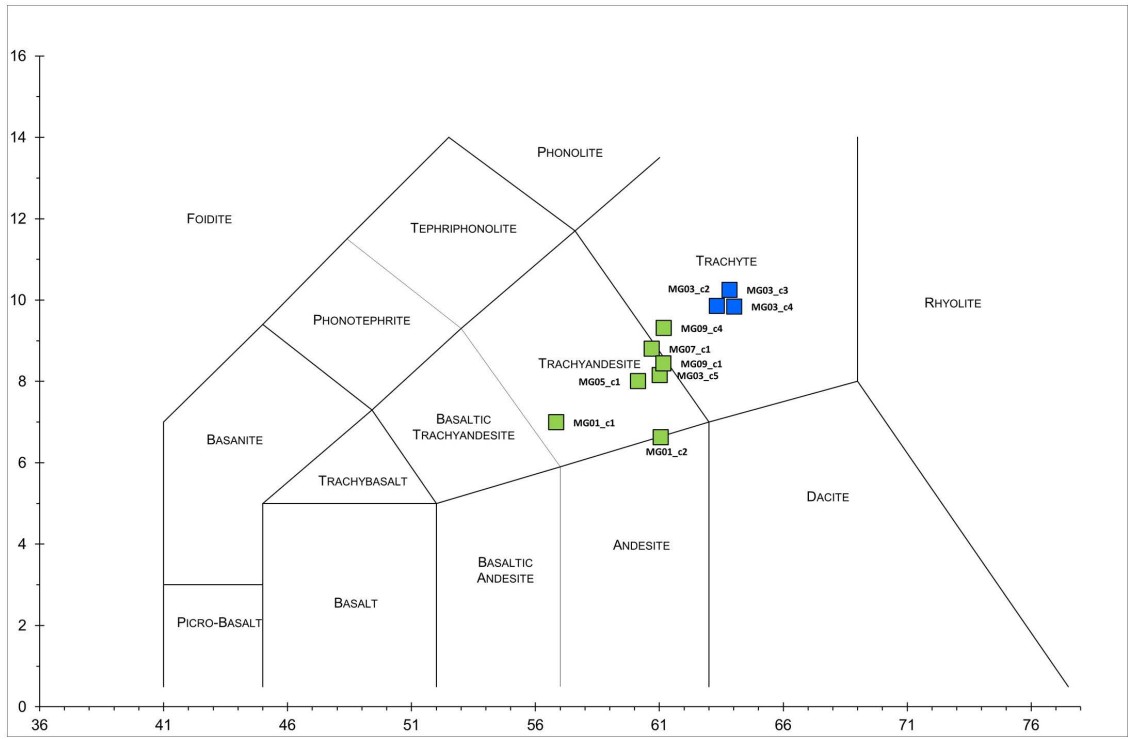

**Fig 6. TAS (Total Alkali Silica) diagram of selected clast samples removed from the mortars and analysed by XRF.** Legend: in blue = samples of the T-group; in green = samples of the T-TA group.

**Table 4. Results of the QPA-XRPD analysis of selected clasts from mortar samples and quarry-correlated samples.** b.d. = below detection limit; * = intrusive component from the mortar; ** = environmental alteration.

| Origin | Sample | | Pla-gioclase | Anortho-clase | Sani-dine | Quartz | Bio-tite | Amphi-bole | Clinopy-roxene | Ilmen-ite | Mag-netite | Smec-tite | Amor-phous | Cal-cite* | Gyp-sum** |
|---|---|---|---|---|---|---|---|---|---|---|---|---|---|---|---|
| Via Scavi mortars | **MG_01-c1** | Clast in mortar | 28,2 | 3,3 | 2,1 | 4,5* | 0,9 | 2,2 | b.d. | 0,2 | 1,0 | 3,9 | 49,8 | 3,9 | b.d. |
| | **MG_03-c5** | Clast in mortar | 43,7 | 13,9 | 1,5 | 0,8 | 0,7 | 2,0 | 0,7 | 0,1 | b.d. | 7,2 | 27,7 | 1,6 | b.d. |
| Villa Draghi quarry | **VD_01-c1** | Clast in breccia | 28,0 | 2,6 | 8,3 | 0,5 | 2,1 | 1,3 | 1,1 | 0,6 | 0,9 | 24,4 | 30,0 | b.d. | b.d. |
| | **VD_02-c2** | Clast in breccia | 31,0 | 2,0 | 9,7 | 0,6 | 2,0 | 1,3 | 0,5 | 0,6 | 0,8 | 16,2 | 34,2 | b.d. | 1,2 |
| Via Scagliara quarry | **MCA_14-c1** | Clast in breccia | 28,1 | 4,0 | 4,8 | 1,0 | 1,2 | 2,1 | 1,2 | 0,6 | 0,5 | 8,3 | 48,3 | b.d. | b.d. |
| | **MCA_16-c2** | Clast in breccia | 34,2 | 22,2 | 0,3 | 0,6 | 0,8 | 1,5 | 2,3 | 1,1 | 0,5 | 11,5 | 25,1 | b.d. | b.d. |

For reliable provenance tracking, the selected clasts were compared with the geochemical fingerprint of the Euganean volcanic rocks in the UNIPD reference database. Only trachytic and trachyandesitic compositions (specifically trachytes and latites) were considered, thereby excluding Euganean rhyolites and basalts outcropping in the Euganean District.

An initial screening of trace element data using matrix diagrams revealed that the clasts of the T-TA group exhibit Sr concentrations > 650 ppm. These values exceed the typical Sr range for Euganean trachytic and latitic rocks, which is ranging around 40–650 ppm. In addition, these clasts show unusual high concentrations of Nb, Ba, Zr, and Nd across all measured chemical variables.

Therefore, trace element data for Ba, Nb, Nd, and Zr were plotted against Sr in tailored bivariate discriminant diagrams (Fig 7 and Fig 8). Across all diagrams, the clasts of the T-TA group consistently show a unique geochemical overlap with

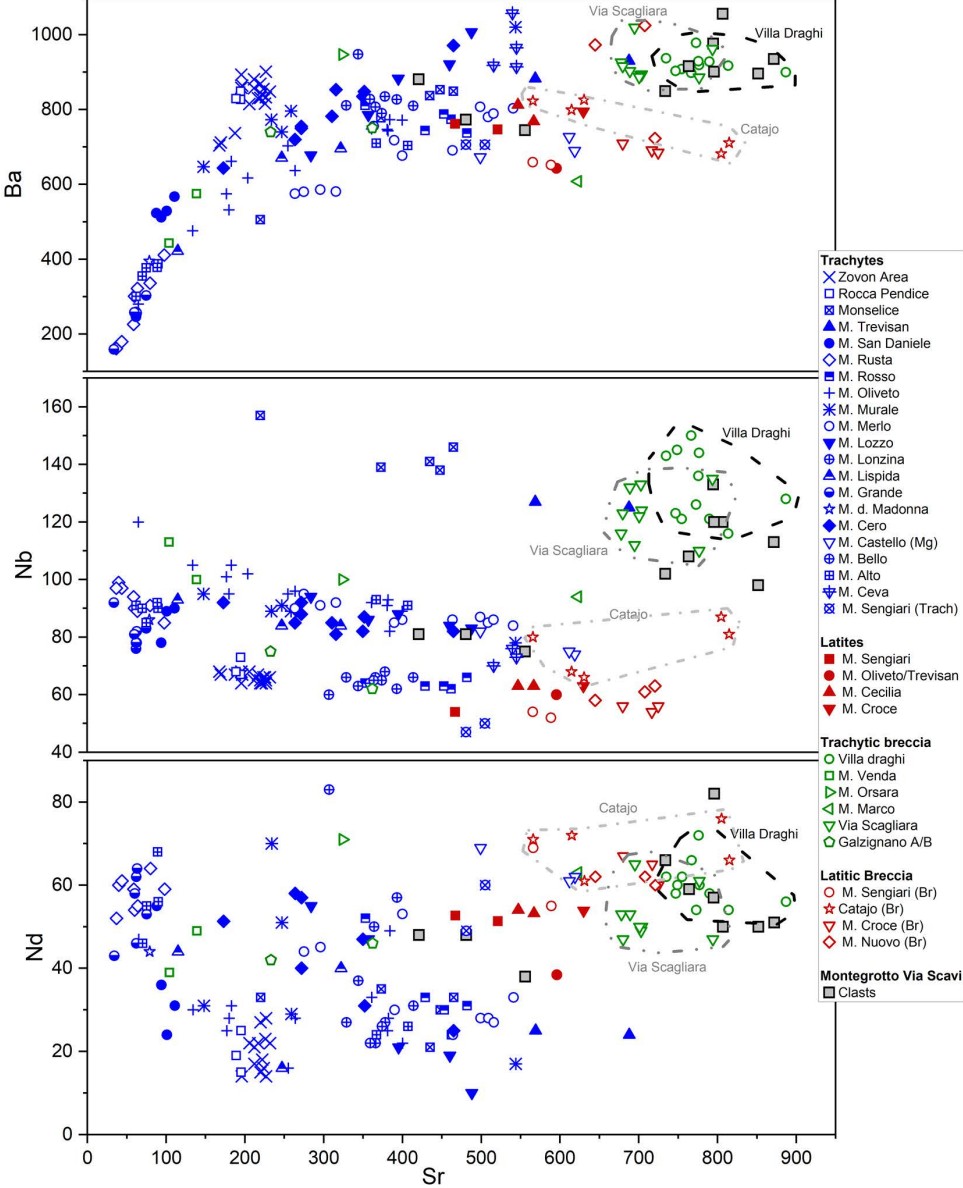

**Fig 7. Sr vs Ba, Sr vs Nb and Sr vs Nd scatterplots used for provenance discrimination of the archaeological volcanic rock clasts from mortars of Via Scavi in Montegrotto Terme (geological marker samples plotted from the UNIPD reference database).** The reference clusters with homogeneous samples from the same quarry sites are marked by dashed or dotted and dashed lines. Certain archaeological samples are labelled and described in detail in the main text.

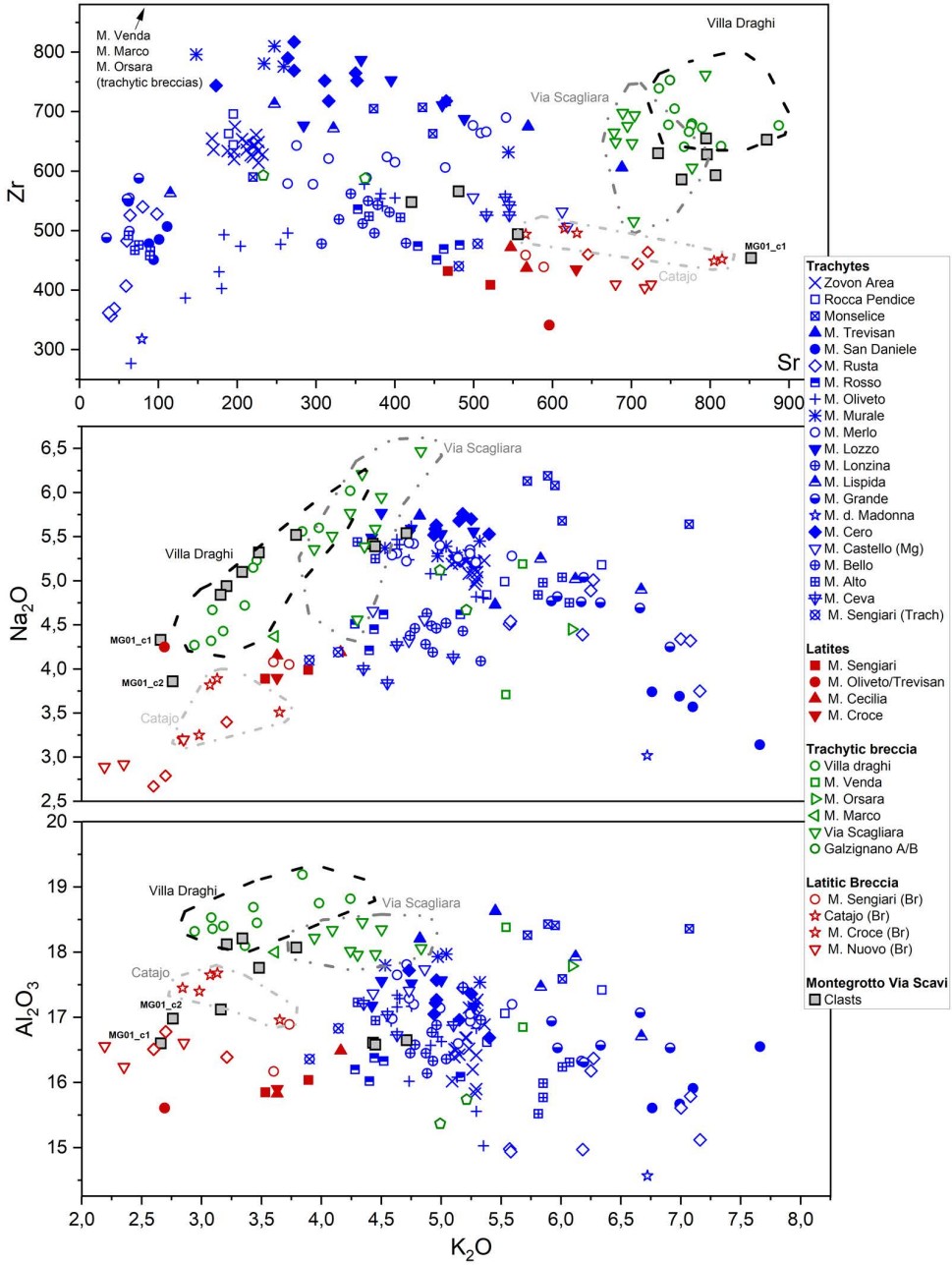

**Fig 8. K₂O vs Zr, K₂O vs Na₂O, K₂O vs Al₂O₃ scatterplots used for provenance discrimination of the archaeological volcanic rock clasts from mortars of Via Scavi in Montegrotto Terme (geological marker samples plotted from the UNIPD reference database).** The reference clusters with homogeneous samples from the same quarry sites are marked by dashed or dotted and dashed lines. Certain archaeological samples are labelled and described in detail in the main text.

the explosive breccias of the quarries in Villa Draghi (eastern Euganean hills, 45°19'44"N 11°46'09"E) and Via Scagliara, north of Mount (hereafter defined M.) Castellone, south-eastern Euganean Hills (45°18'46"N 11°46'00"E) and recently pinpointed using UAV-LiDAR-based surveys [48] (see Fig. 1, c1 and c2). The reference geological samples from these

two outcrops yield Sr concentrations ranging from 678 to 794 ppm at Via Scagliara quarry and from 735 to 887 ppm at Villa Draghi. Compared with more typical Euganean trachytes, the higher contents of these elements is consistent with the rapid disequilibrium in crystallization of alkali feldspar during explosive events, which enhances the incorporation of Sr and Ba into the solid phase [63]. The Villa Draghi and Via Scagliara breccias also display comparable Nb concentrations, while most of the other Euganean trachytes, latites, and breccias exhibit lower Nb values (<100 ppm), except for samples from the Monselice and M. Trevisan quarries (~140–150 ppm).

Further scatterplots of major elements support this interpretation, suggesting that specific major oxide compositions can also serve as effective markers for distinguishing Euganean breccias from their lava counterparts. In fact, clasts of the T-TA group are notably depleted in $Na_2O$ and $K_2O$ in comparison with most Euganean trachytes and latites, while showing a slight enrichment in $Al_2O_3$ compared to other Euganean volcanic rocks (see Fig 8). Again, a strong geochemical match is observed with breccia samples from Villa Draghi and Via Scagliara, with Villa Draghi samples exhibiting slightly lower $K_2O$ concentrations (<4.5%ox.) compared to Via Scagliara ($K_2O$~3.8–5.0%ox.). The most altered clast samples, MG_01-c1 and MG_01-c2, stand out due to their markedly depleted $Na_2O$, $K_2O$, and $Al_2O_3$ contents that likely reflect pozzolanic reactions and early-stage mobilization of light elements (e.g., Na) through interaction with the lime binder.

These results strongly indicate that the clasts of the T-TA group likely originated from either Villa Draghi or Via Scagliara quarry sites. Other possible quarries of Euganean volcanic breccia exhibiting trachytic-latitic chemical composition, as those from M. Marco, M. Orsara, M. Venda (trachytic clasts in a primarily rhyolitic breccia facies), the area of Galzignano (trachytic breccia) (see Fig 1, f) do not return geochemical compatibility. Similarly, the latitic breccias from M. Sengiari (see Fig 1, b), M. Croce (see Fig 1, e), Turri north of M. Nuovo and in the area around the Catajo Castle (southern edge of M. Ceva) (see Fig 1, d), recently identified through UAV-LiDAR surveys [48], show only occasional or marginal geochemical overlap in scatterplots and can therefore be ruled out as probable sources.

In contrast, the T-group clasts plot squarely within the Euganean trachyte field in all relevant major- and trace-element diagrams and can therefore be confidently identified as fragments of Euganean trachyte lavas. Therefore, these are the clasts that exhibited no reaction rims with the binder after TPL-OM analyses. Precise quarry provenance of such clasts was not pursued as it lies beyond the scope of this study.

Further analyses via TPL-OM reinforce the compatibility of the T-TA group with the identified quarries. Overall, these rocks exhibit textural characteristics and rock-forming mineral assemblages (Fig 9 a–h) that are substantially identical to those observed in the archaeological samples (see S2 Table for detailed mineral compositions). These features are typical of chaotic explosive diatreme breccias [36]. Although the brecciated texture is not always visually evident in thin section, this is simply due to the large size of the textural domains relative to the average size of the individual fragments within the matrix.

The coarse clasts MG_01-c1 and MG_03-c5 were also analysed via QPA-XRPD and compared with Villa Draghi and Via Scagliara reference samples (S1 Fig). These materials exhibit a coherent mineral assemblage aligning with petrographic observations (Table 4), providing additional mineralogical evidence of compatibility. Crucially, certain features clearly distinguish these samples from the typical Euganean effusive-subvolcanic rocks. For instance, neither cristobalite nor tridymite, phases frequently found in Euganean trachytes [41,49,51] and generally indicative of slow cooling at high temperatures, was detected in any of the samples. Moreover, the most distinguishing feature is the substantial amorphous fraction (25.1–49.8%wt.). This is attributable to the relevant glassy matrix, which reflects the textural and compositional heterogeneities typical of volcanic breccias, where rapid and spatially variable cooling regimes exert a strong control on crystallization kinetics, mineral assemblages, and modal proportions. Ultimately, this high amorphous content allows for the unambiguous discrimination of these explosive products from the Euganean effusive-subvolcanic rocks, which systematically show much lower values (≤20%wt. under identical conditions) [49,51]. Other mineral phases appear not discriminant. While minor rock-forming minerals are generally present at low concentrations in both the quarry and archaeological samples, sanidine content varies, with the Villa Draghi quarry samples displaying higher amounts (5–10%wt.). Quartz is typically <1.0%wt., except in the archaeological sample MG_01-c1 (>4.0%wt.), where it is interpreted as mortar contamination.

**Fig 9. Volcanic diatreme breccia samples from Villa Draghi and Via Scagliara quarries.** (a) fresh crosscut of a representative sample from Villa Draghi quarry; (b-d) micrographs acquired in TPL-OM (XP, PP) of representative portions of the Villa Draghi breccia samples; (e) fresh crosscut of a representative sample from Via Scagliara quarry; (f-h) micrographs acquired in TPL-OM (XP, PP) of representative portions of the Via Scagliara breccia samples. Legend: Bc = breccia clasts; Bt = biotite; Gm = groundmass; Ks = kaersutite (amphibole); Op = opaque minerals; Pl = plagioclase.

Nevertheless, due to their strikingly similar characteristics, the Villa Draghi and Via Scagliara samples exhibit a high degree of petro-mineralogical uniformity, making them indistinguishable by TPL-OM and QPA-XRPD and preventing any further discrimination between the two outcrops based solely on these observations.

## Provenance discrimination between the Villa Draghi and Via Scagliara outcrops

The ultimate goal of the protocol was to discriminate between Villa Draghi and Via Scagliara as the most likely source quarry. To address this target, a series of tailored bivariate scatterplots was produced using trace elements and selected after testing the full dataset of XRF data in matrix diagrams. While thorium (Th) showed slightly higher concentrations in Via Scagliara (27–34 ppm) than in Villa Draghi (23–28 ppm), plotting T-TA clasts against other discriminant elements (Sr, Nb, Ga, V) revealed intermediate compositions and substantial overlap between the two reference ranges (S2 Fig), indicating that these elements alone do not allow clear source discrimination.

The XRF data were also processed using Linear Discriminant Analysis (LDA). The model incorporated a selected subset of major ($Na_2O$, $Al_2O_3$, $K_2O$) and trace elements (V, Sr, Zr, Nb, Ba, Nd, Th) previously validated through bivariate exploration in matrix diagrams. Gallium (Ga) was excluded from the analysis as its concentrations fell below the detection limit in specific clasts (MG_05-c1 and MG_01-c2).

The LDA achieved a 100% correct classification rate for the 22 geological reference samples (Villa Draghi, n = 13; Via Scagliara, n = 9), testing both the "full variable" and "stepwise forward" predictive models. Considering the small statistical representativeness of the training dataset, the robustness of the model has been validated via Leave-One-Out Cross-Validation (LOOCV). The LOOCV yielded a 100% correct classification rate for both the Full and Stepwise models, confirming the effective discriminant power of the selected markers (Table 5). When the predictive model was applied to the

archaeological volcanic aggregates of the T-TA group, the majority of the clasts exhibited a high posterior probability of originating from Villa Draghi. Nonetheless, the LDA "full variable" model assigned clasts MG_09-c4 and MG_05-c1 to the Via Scagliara outcrop with 99.4% and 99.1% probabilities, respectively (Table 6).

However, despite the high statistical accuracy, the limitations of Linear Discriminant Analysis (LDA) must be taken into account. As a supervised technique, LDA operates under a 'closed-world assumption,' forcing every unknown sample into one of the predefined groups. Furthermore, the classification proved highly sensitive to the specific model configuration [64]. A comparative evaluation revealed that the "full variable" model (Table 6) and a "stepwise forward" model (Table 7) yielded conflicting provenance outcomes for "ambiguous" samples MG_09-c4 and MG_05-c1, but it must be outlined that the observed discrepancies stem more from material-specific geochemical ambiguities than from pure statistical model-ling. The limited size and heterogeneity of the clasts, combined with the potential pozzolanic alterations at the interface, can deviate geochemical signatures from their geogenic and authigenic fingerprint, particularly when dealing with slight fluctuations in elemental concentrations. Consequently, LDA probabilistic outputs reflect the sensitivity of transitional

**Table 5. LDA classification matrix and model validation. The model correctly classified 100% of the training set (Full analysis and Stepwise Forward models). Classification was performed using equal prior probabilities (0.50) to avoid bias toward either source.**

| Actual Provenance | Group Size (n) | Predicted: Via Scagliara | Predicted: Villa Draghi | % Correct |
|---|---|---|---|---|
| Via Scagliara | 9 | 9 | 0 | 100 |
| Villa Draghi | 13 | 0 | 13 | 100 |
| Total | 22 | | | 100 |

**Table 6. Results of the Linear Discriminant Analysis (LDA) of volcanic clast samples (Full analysis model), showing their probabilistic assignment to sources between the Villa Draghi and Via Scagliara di M. Castellone quarries. The discriminant function with a p-value < 0,05 is considered statistically significant at the 95% confidence level.**

| Sample | 1st max. Group | 1st Max. value | Squared distance | Probability (%) | 2nd max. Group | 2nd Max. value | Squared distance | Probability (%) |
|---|---|---|---|---|---|---|---|---|
| MG_01-c1 | Villa Draghi | 1640,95 | 109,082 | 100 | Via Scagliara | 3075,49 | 43,1343 | null |
| MG_01-c2 | Villa Draghi | 2186,4 | 7,39798 | 100 | Via Scagliara | 2157,34 | 65,5257 | null |
| MG_03-c5 | Villa Draghi | 2458,5 | 0,183737 | 100 | Via Scagliara | 4406,23 | 15,5102 | null |
| MG_05-c1 | Via Scagliara | 2433,46 | 3,25852 | 99,1 | Villa Draghi | 2428,71 | 12,743 | 0,9 |
| MG_07-c1 | Villa Draghi | 2407,37 | 0,0003477 | 100 | Via Scagliara | 4262,63 | 18,0319 | 2,6 |
| MG_09-c4 | Via Scagliara | 2533,07 | 3,04917 | 99,4 | Villa Draghi | 4185,68 | 79,3185 | 0,6 |
| MG_09-c1 | Villa Draghi | 2555,24 | 0,0710432 | 100 | Via Scagliara | 4744,28 | 51,8505 | null |

**Table 7. Results of the Linear Discriminant Analysis (LDA) of volcanic clast samples (Stepwise Forward model), showing their probabilistic assignment to sources between the Villa Draghi and Via Scagliara di M. Castellone quarries. The discriminant function with a p-value < 0,05 is considered statistically significant at the 95% confidence level.**

| Sample | 1st max. Group | 1st Max. value | Squared distance | Probability (%) | 2nd max. Group | 2nd Max. value | Squared distance | Probability (%) |
|---|---|---|---|---|---|---|---|---|
| MG_01-c1 | Villa Draghi | 262,749 | 0,159214 | 100 | Via Scagliara | 250,09 | 25,4778 | null |
| MG_01-c2 | Villa Draghi | 1,66831 | 167,169 | 100 | Via Scagliara | −69,2389 | 308,984 | null |
| MG_03-c5 | Villa Draghi | 250,416 | 0,004705 | 100 | Via Scagliara | 239,293 | 22,2513 | null |
| MG_05-c1 | Villa Draghi | 285,659 | 2,04507 | 98,5 | Via Scagliara | 281,502 | 10,3586 | 1,5 |
| MG_07-c1 | Villa Draghi | 253,503 | 1,12506 | 100 | Via Scagliara | 237,768 | 32,5952 | null |
| MG_09-c4 | Villa Draghi | 298,024 | 5,08944 | 57,9 | Via Scagliara | 297,707 | 5,72433 | 42,1 |
| MG_09-c1 | Villa Draghi | 207,564 | 10,6497 | 100 | Via Scagliara | 181,59 | 62,5985 | null |

signatures to different statistical configurations and it must therefore be interpreted alongside additional analytical and archaeological constraints rather than just as a definitive provenance tool.

To overcome the limitations of bulk XRF geochemistry and critically evaluate the provenance link between Via Scagliara and Villa Draghi, we focused on individual mineral phases. Specifically, we selected minerals whose crystallographic properties make them non-reactive toward the lime binder, ensuring that their geochemical signatures remain reliable tracers of the original Euganean lithotypes [50]. The phases investigated included plagioclase (the dominant mineral), along with biotite, brown amphibole, and magnetite, identified in both quarry samples and reacted volcanic clasts (breccias) from the mortars. Although advanced techniques such as (LA)ICP-MS or EMPA are commonly employed in provenance studies for crystal-based discrimination, in this study, where the objective was restricted to distinguishing between two specific outcrops, SEM microstructural analyses and EDS calibrated as microprobe for quantitative estimation (S1 File) provided adequate resolution, making the application of more sophisticated methods unnecessary. Microchemical data and classification diagrams for plagioclase, amphibole, biotite, and magnetite are reported in the S2 Table. The biotite formula was calculated following [65,66], and compositions were plotted in diagrams after [67]. The amphibole formula was calculated using a spreadsheet developed by [68], based on the recommendation of [69]. The magnetite formula was calculated using OxyEMG software [70,71], and data projected in the genetic discrimination diagrams proposed by [72–74].

Plagioclase, amphibole, and biotite showed no significant chemical or microstructural variability, nor compositional differences between samples from the two quarries, and from the breccia clasts of the mortars (S2 Table), thus they resulted not useful for ultimate provenance discrimination. By contrast, magnetite yielded clear diagnostic features. We observed a distinctive microstructure of early magnetite partially replaced by a Si-rich magnetite with a colloform-banding texture – similar to what observed by [75] in the Chilean Mariela iron oxide-apatite deposit and by [76] in the Shepherd Mountain iron ore deposit in Southeast Missouri, USA, in both the mortars' clasts (Fig 10) and Villa Draghi quarry samples (Fig 11). As observed via targeted ESD chemical mapping, magnetite exhibits clear Si-enrichments in the regions connoted with a colloform-banding texture, which appear Ti- and Fe-depleted (Fig 12), with $SiO_2$ concentration usually exceeding 2%ox. and even reaching ~8%ox. Standardized EDS analyses in these areas frequently yielded low analytical totals of elemental concentrations, often well below 100%ox. (in some cases, ~80%ox.), suggesting the presence of hydrated phases and micropores formed during secondary transformations after magnetite. Furthermore, μ-Raman spectra revealed the presence of magnetite, sometimes partially maghemitized (S3 Fig). The microstructure and compositional patterns of the Si-rich, colloform-banded rims – notably their nucleation at magnetite margins and continuation into internal fractures, together with maghemitization – are most consistent with low-temperature hydrothermal replacement of magmatic magnetite [77,78]. By contrast, magnetite from Via Scagliara is texturally fresh (Fig 11), and shows consistent $SiO_2$ content ≤ 0.2%wt., reflecting unaltered primary magmatic crystals. The lack of this distinctive mineralogical marker in the mortars' clasts from Via Scavi provides an unambiguous fingerprint, pinpointing Villa Draghi as the unique provenance for the breccias and confirming that the supply of pozzolanic material required for the construction of the Via Scavi Roman complex was drawn entirely from this specific outcrop.

## Archaeological inferences and broader implications of material circulation

Archaeometric analyses have demonstrated the use of volcanic pozzolans extracted from the Euganean Hills during Roman times, revealing the exploitation of a raw material previously undocumented in both archival sources and scientific literature. These materials were sourced from explosive-diatreme breccia outcrops – many previously unknown, overgrown, or misidentified – here recognized for the very first time through comparison of the clasts in the archaeological mortars with Euganean volcanic samples from multiple quarry sites. Among these, the Villa Draghi quarry, named after a nearby 19th-century villa, lies on the eastern edge of the Euganean Hills, opposite the trachytic outcrop of M. Castello. Evidence of both open-air and tunnel manual extraction, with visible ~45° angle pick marks on the quarry faces, prove the exploitation of this site during pre-industrial times (Fig 13, a–b) [79]. UAV-LiDAR mapping combined with ground

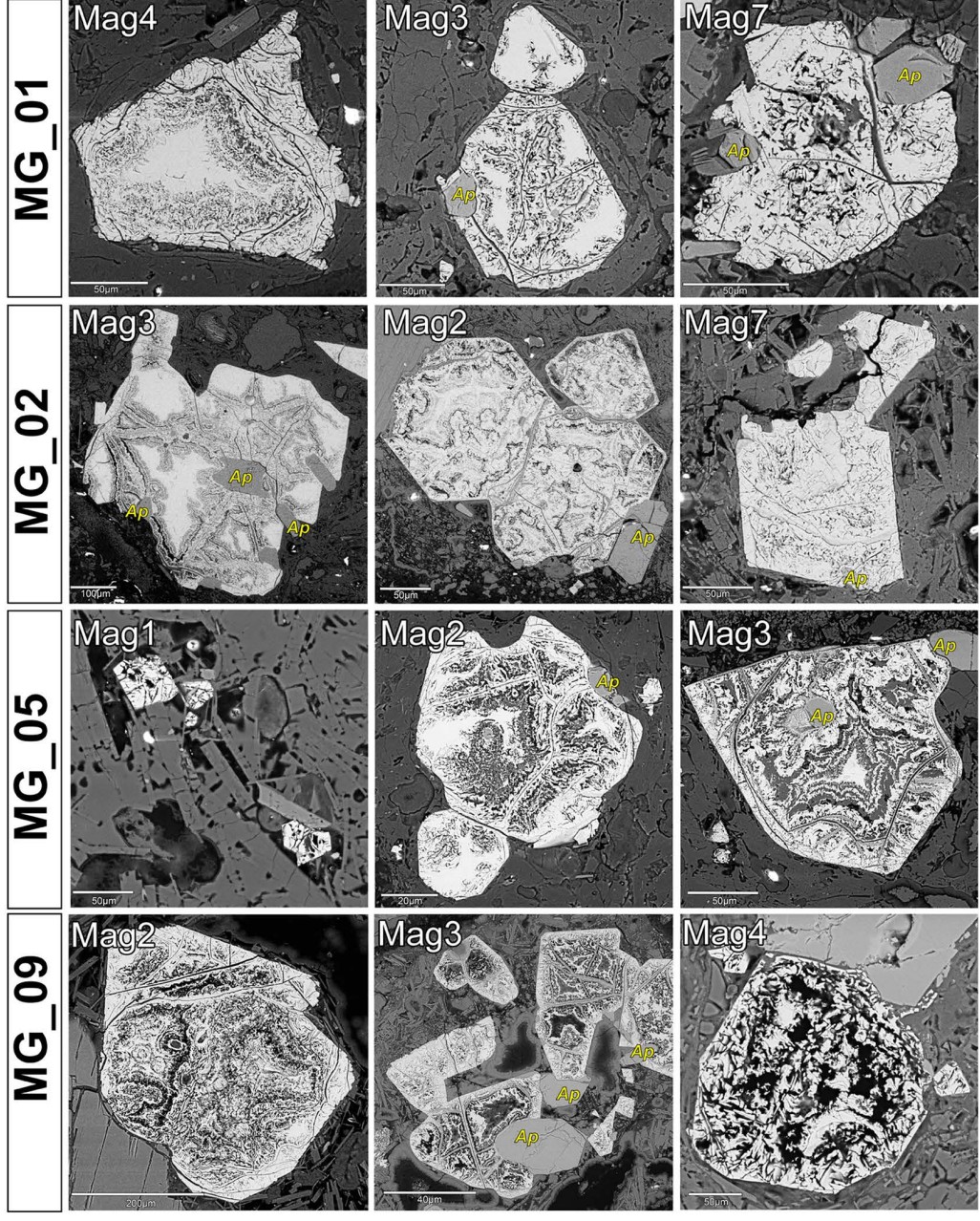

**Fig 10. SEM-BSE images of magnetites from breccia clasts in selected mortar samples MG_01, MG_02, MG_05, MG_09.** Legend of minerals other than magnetites: Ap = apatite.

survey enabled the reconstruction of approximately 700 m of quarry front, with an estimated minimum extracted volume of approximately 27,000 m³ of material (Fig 13, c). Similar traces of manual extraction were also documented at the area of Via Scagliara di M. Castellone (Fig 13, d), with an estimated minimum extracted volume of approximately 7,700 m³ of material (Fig 13, e), thus confirming both localities as plausible sources of pozzolanic materials for construction [48]. Notably, these quarries are absent from 20th century cartographic records of historical Euganean quarries [47], which underscores the extent to which their exploitation had been forgotten.

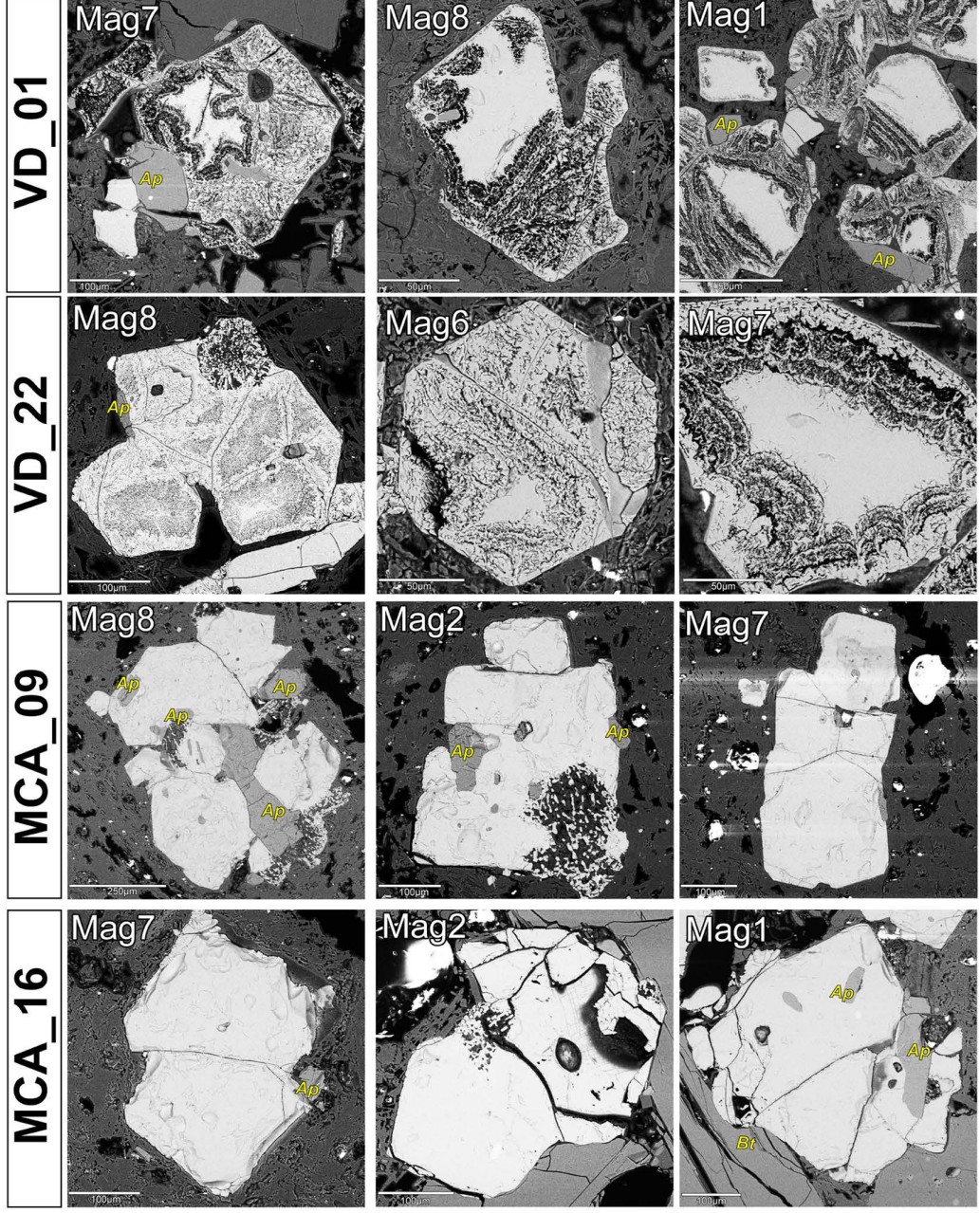

**Fig 11. SEM-BSE images of magnetites from breccia clasts in selected geological samples from Villa Draghi (VD_01, VD_22) and Via Scagliara of M. Castellone (MCA_09, MCA_16).** Legend of minerals other than magnetites: Ap = apatite; Bt = biotite.

This discovery highlights the intimate bond between the Roman builders and their territory. The targeted exploitation of these minor outcrops provides original insight into the advanced empirical knowledge on material properties of Roman crafts and it highlights how a sophisticated ability to recognize geological features enabled builders to select materials ideally suited to their construction needs: trachytes and rhyolites, being relatively compact and durable [41], were preferentially used for masonry blocks, architectural elements, paving stones (e.g., at Montegrotto), as well as for specific artifacts

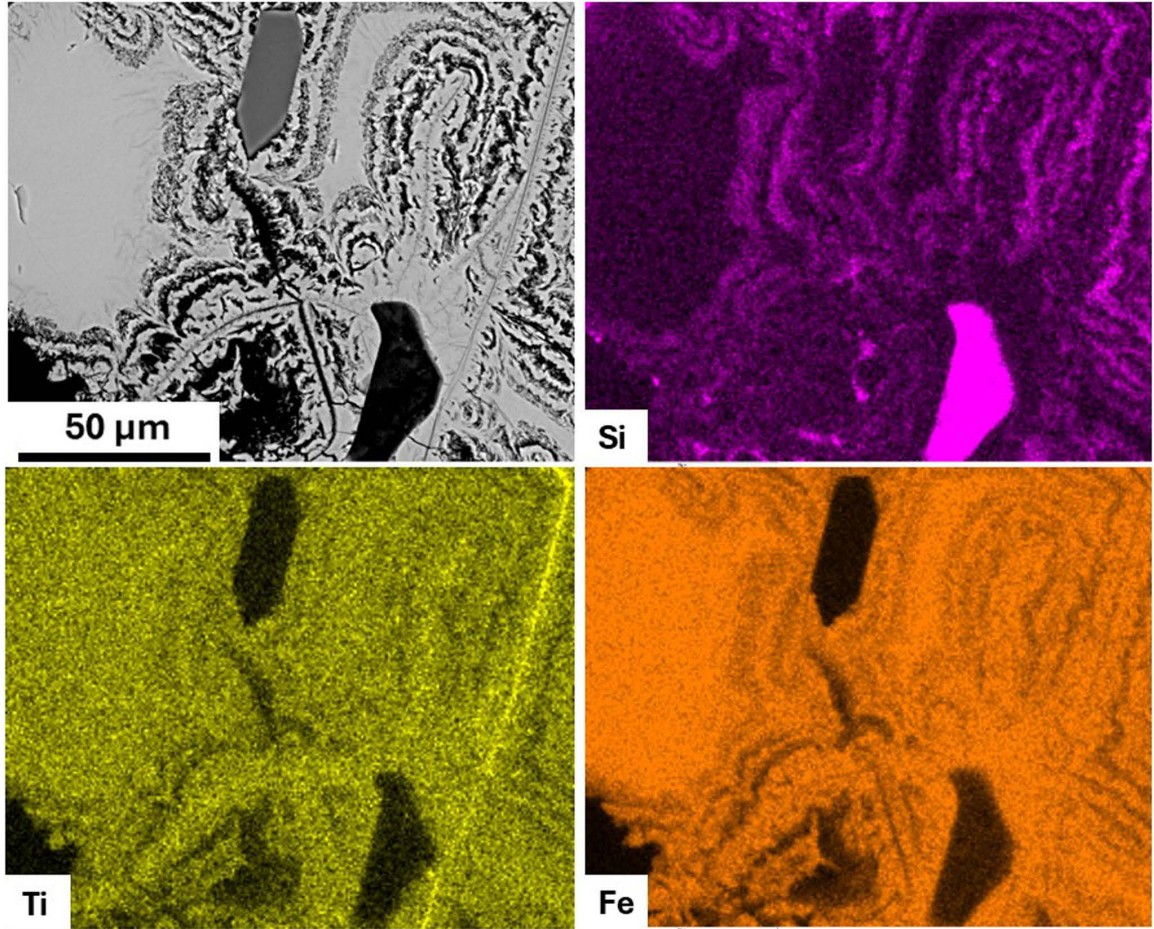

**Fig 12. EDS chemical mapping of magnetite from Villa Draghi geological samples (VD_02).**

such as millstones, and gravestones. In contrast, the trachytic breccias appear to have been employed almost exclusively as aggregates for mortars' production. Their selection was likely guided by the material's macroscopic affinity to volcanic tuffs and its inherent low hardness, which facilitated efficient quarrying and comminution while ensuring superior mechanical and hydraulic performance in the final composites.

Indeed, one must not be misled by the mere geographical proximity of these georesources to the site of destination. In fact, at first glance, the proximity of the Villa Draghi quarry to Montegrotto Terme (distance: ca. 1.8 km), might suggest just an opportunistic exploitation of the nearest available resource for the construction of the Bath-Theatre complex at Via Scavi.

However, recent archaeological data demonstrate that the use of "Euganean pozzolans" was not confined to local contexts and may have extended well beyond regional boundaries. Based on preliminary petrological, mineralogical and geochemical analyses, Rubinich et al. [80] suggested that some volcanic clasts exhibiting pozzolanic reactions within the *opus caementicium* foundations of the Great Baths of Aquileia – located approximately 150 km northeast of the Euganean District – are consistent with Euganean volcanic lithotypes, even though their specific provenance was not pinpointed. The reprocessing of these data against the updated UNIPD geochemical reference database confirms that these clasts share the same discriminant markers as those in the mortars from Via Scavi. In all discriminant diagrams (S4 Fig and S5 Fig), one clast we previously analysed via XRF (adopting the same equipment described in S1 File) consistently overlap

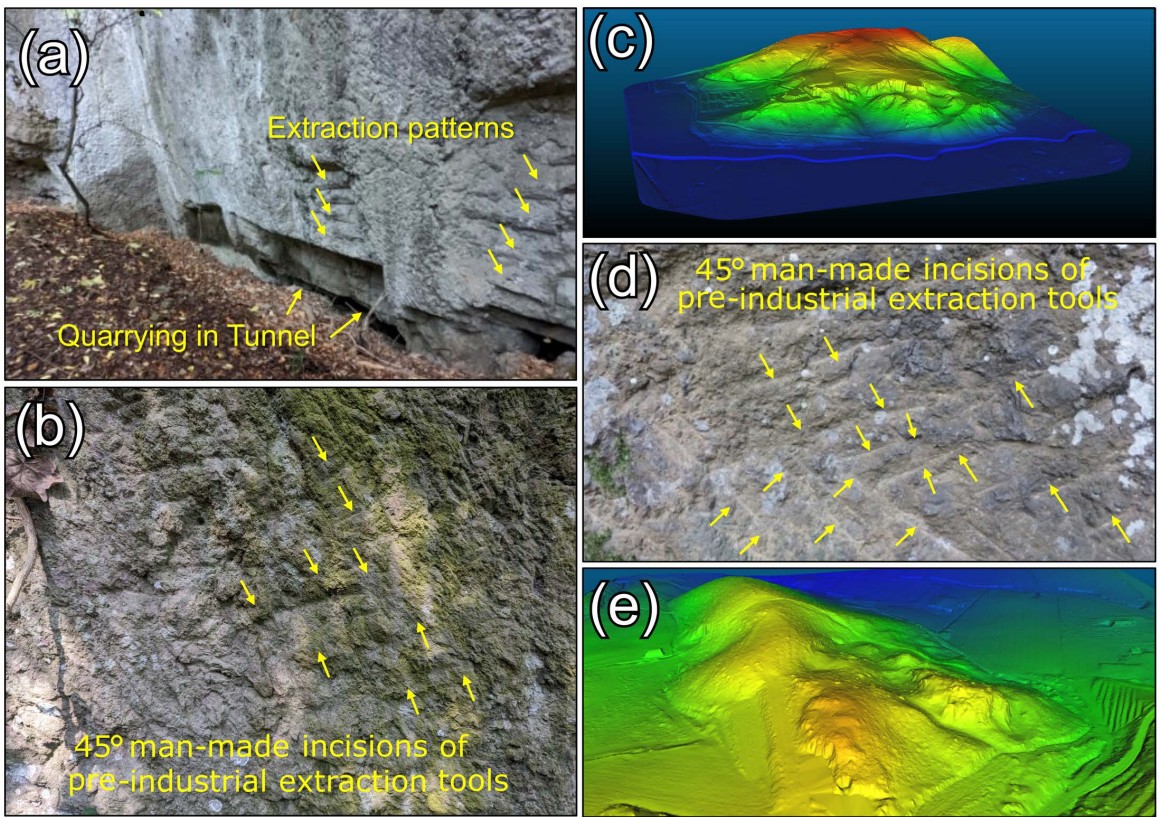

**Fig 13. The Villa Draghi and Via Scagliara quarries. (a-b)** Manual extraction traces, consisting in 45° incisions let by picks at Villa Draghi and evidence of tunnel quarrying; **(c)** UAV-LiDAR based reconstruction of the Villa Draghi quarry front (from [48]); **(d)** Manual extraction traces, consisting in 45° incisions let by picks at Via Scagliara quarry; **(e)** UAV-LiDAR based reconstruction of the Via Scagliara quarry front (from [48]).

the compositional fields of the Villa Draghi and Via Scagliara breccias, confirming that "Euganean pozzolans" were used in Aquileia and circulated across substantial distances. Moreover, SEM-EDS analyses of magnetite within the volcanic breccias in mortars from the Great Baths of Aquileia revealed the distinctive Si-rich colloform-banding textures (S6 Fig) characteristic of the Villa Draghi quarry samples, thereby identifying this site as the most likely source of the material.

In the broader context of Roman construction materials' supply networks, the importation of Euganean volcanic rocks to Aquileia is not a new finding. Trachyte lavas, for instance, were widely employed for paving during the early Imperial period [40,81], as confirmed by petrographic analyses [82,83]. Trachyte millstones preserved in the Archaeological Museum of Aquileia have also been traced to Euganean sources through multi-analytical methods [84]. However, the available evidence broadens this perspective by showing that Euganean volcanic breccias were also exploited and circulated as pozzolanic materials on an extra regional level. Although the data for Aquileia remain preliminary and are based on a statistically limited dataset, the identification of "Euganean pozzolans" in foundation contexts – clearly employed for functional purposes (e.g., enhancing the mechanical strength of the foundational *opus caementicium*) – within a large public building of imperial patronage strongly points to a deliberate and well-planned supply strategy. This finding implies that the movement of volcanic resources into north-eastern Italy was not confined to well-known lithotypes such as trachyte but instead formed part of a broader and more complex network of material procurement and mobility. Significantly, while the Montegrotto contexts date between the 1st and 2nd centuries CE, the Aquileia evidence belongs to the late 3rd-early 4th

century CE [80]. This temporal shift indicates that the extraction and utilization of Euganean breccias for mortar production was not a sporadic or short-lived activity, but rather a protracted practice that endured for at least three centuries.

In conclusion, this discovery on the use of "Euganean pozzolans" pose significant historical, socio-economic, and archaeological questions about the organization of construction materials supply systems, the scale of regional and trans-regional trade in building materials, and the technical expertise required to identify and exploit suitable volcanic resources. Addressing this issue, future research should prioritize interdisciplinary frameworks that consciously embed archaeometric evidence within economic, social, and technological narratives to produce historically grounded reconstructions of Roman construction systems, trade networks, and production economy.

## Conclusions and main methodological outcomes

This study presents the identification of a new type of volcanic pozzolan, distinct from the canonical Vitruvian sources of central and southern Italy, and explores the extent of technological originality beyond the traditional narrative. These results underscore the advanced empirical knowledge that ancient craftsmen possessed regarding local geological resources and their suitability for mortar-based construction. Beyond the specific case studies (Montegrotto Terme and Aquileia) the research also offers methodological contributions of broader significance:

- **Accessible analytical strategies:** This study demonstrates that combining relatively rapid techniques (petrographic, mineralogical, and bulk geochemical analyses) effectively supports provenance attributions. By progressively ruling out incompatible sources, we reliably discriminated between potential quarries using a tailored, stepwise protocol: (i) Geochemical screening (XRF database comparison)> (ii) Validation of petrographic and mineralogical features of compatible samples (via TPL-OM and QPA-XRPD)> (iii) Mineral microchemical investigation (SEM-EDS, primarily on magnetite) for final discrimination of the two sources. This iterative strategy improves the resolution and robustness of provenance assessments, providing a practical and reproducible framework for future archaeometric research.

- **Comprehensive reference databases:** Provenance studies, at both local and regional scales, must rely on systematically compiled reference databases encompassing all potential source areas. In this study, the availability of a dedicated reference dataset for the Euganean Hills Magmatic District, including both geological samples and archaeological comparands probed with the same analytical techniques, enhances the reliability, comparability, and internal coherence of provenance assignments.

- **Quarry-level provenance insights:** This study traces volcanic pozzolans to a specific quarry, advancing beyond the broader district-level attributions common in previous work. The identification of an extraction site likely abandoned after the Roman period [48] suggests that ancient builders detected and exploited pozzolan sources with a spatial resolution on the order of tens of square meters. These findings reveal an incredibly sophisticated grasp of local geology from two centuries ago: the crafts' ability to recognize and selectively exploit small outcrops for specific uses, such as pozzolanic aggregates for mortars, demonstrates an impressive level of technical and territorial mastery. Paradoxically, this prompts a reassessment of our modern methods: despite our advanced technology and theoretical expertise, contemporary practice has largely lost touch with the profound, direct empirical knowledge of materials that humans from the ancient past possessed.

## Supporting information

**S1 Fig. XRPD patterns of the breccia clasts from mortar samples and quarry samples of Villa Draghi and Via Scagliara di M. Castellone.** Mineral phases are labelled according to [61] (when mentioned): Sme = smectite; Bt = biotite; Amp = amphibole (horneblende type); Gp = gypsum; Sa = sanidine; Qz = quartz; Ano = anorthoclase; Pl = plagioclase; Cal = calcite; Cpx = clinopyroxene; Ilm = ilmenite; Mag = magnetite; Znc = zincite (internal standard).
(DOCX)

**S1 File. Instrumental equipment and standards.**
(DOCX)

**S1 Table. Results of XRF analyses on volcanic clasts extracted from mortar samples of the archaeological site of Via Scavi in Montegrotto Terme.** b.d. = below detection limit.
(XLSX)

**S2 Fig. Selected discriminant provenance scatterplots for Villa Draghi and Via Scagliara di M. Castellone quarries.** The archaeological clasts exhibit intermediate distribution among the quarry intervals.
(DOCX)

**S2 Table. Results of SEM-EDS analyses on main minerals (amphibole, biotite, plagioclase, magnetite) within breccia clasts from quarry sites of Villa Draghi and Via Scagliara di M.** Castellone and archaeological samples from Via Scavi in Montegrotto. Sheets are related to the four analysed categories (amphibole, biotite, plagioclase, magnetite). Legend: *Samples*: MG_01, MG_02, MG_04, MG_09 = archaeological mortars from Via Scavi; VD_01 and VD_22 = quarry samples from Villa Draghi; MCA_09 and MCA_16 quarry samples from Via Scagliara di M. Castellone; *Grain*: numbering of the mineral within the four analyzed categories (Amp = amphibole; Bt = biotite; Pl = plagioclase; Mag = magnetite). *Spectrum* = punctual EDS analysis.
(XLSX)

**S3 Fig. µ-Raman spectra of magnetite crystals, exhibiting diagnostic peaks attributable to both magnetite and maghemite (maghemitized magnetite).** Sample VD_02 from Villa Draghi Quarry (Spectra 1–4); sample MCA_16 (Spectrum 5).
(DOCX)

**S4 Fig. Sr vs Ba, Sr vs Nb and Sr vs Nd scatterplots used for provenance discrimination of the archaeological volcanic rock clasts from the foundational mortars of the Late Antique Baths of Aquileia.** The reference clusters with homogeneous samples from the same quarry sites are marked by dashed or dotted and dashed lines. Geological marker samples are plotted from the UNIPD reference database.
(DOCX)

**S5 Fig. $K_2O$ vs Zr, $K_2O$ vs $Na_2O$ and $K_2O$ vs $Al_2O_3$ scatterplots used for provenance discrimination of the archaeological volcanic rock clasts from the foundational mortars of the Great Baths of Aquileia.** The reference clusters with homogeneous samples from the same quarry sites are marked by dashed or dotted and dashed lines. Geological marker samples are plotted from the UNIPD reference database.
(DOCX)

**S6 Fig. SEM-EDS analysis of magnetite crystals within the volcanic breccias included in the foundational mortars of the Great Baths of Aquileia.** Sample GTR_S5_M3-c1, described in [80]. Legend: Ap = apatite.
(DOCX)

## Acknowledgments

This work is part of the "Geoarchaeology of Euganean quarrying from research to valorization – EuQuGeA" project. We thank the entire EuQuGeA team for their significant contributions to these results. In addition, acknowledgment is given to Associazione Villa Draghi, specifically Dr. Elvio Cognolato, for their role in the investigation of the Villa Draghi quarry site, as well as to Enrico Lorato for providing entrance permission and support during the investigation of the Via Scagliara quarry site.

Furthermore, we extend our gratitude to the Soprintendenza Archeologia, Belle Arti e Paesaggio per l'area metropolitana di Venezia e le province di Belluno, Padova e Treviso (in particolar Dr. Carla Pirazzini and Dr. Elena Pettenò) for the authorization at sampling activities (MIC_SABAP-VE-MET 23/06/2021 001975-P), and to the Museo del Termalismo Antico e del Territorio and Assocazione Lapis (specifically to Dr. Chiara Destro), for their support and endorsement in the archaeological research activities.

## Author contributions

**Conceptualization:** Simone Dilaria, Luigi Germinario, Michele Secco.

**Data curation:** Simone Dilaria, Luigi Germinario, Claudio Mazzoli, Milo K. Pilgrim, Josiah Olah, Jacopo Nava, Michele Secco.

**Formal analysis:** Simone Dilaria, Luigi Germinario, Claudio Mazzoli, Milo K. Pilgrim, Michele Secco.

**Funding acquisition:** Jacopo Bonetto, Michele Secco.

**Investigation:** Simone Dilaria, Luigi Germinario, Claudio Mazzoli, Jacopo Nava, Michele Secco.

**Methodology:** Simone Dilaria, Luigi Germinario, Claudio Mazzoli, Josiah Olah, Michele Secco.

**Project administration:** Michele Secco.

**Resources:** Michele Secco.

**Software:** Simone Dilaria.

**Supervision:** Jacopo Bonetto, Michele Secco.

**Validation:** Simone Dilaria.

**Visualization:** Simone Dilaria, Claudio Mazzoli, Caterina Previato.

**Writing – original draft:** Simone Dilaria, Luigi Germinario, Claudio Mazzoli, Caterina Previato, Michele Secco.

**Writing – review & editing:** Simone Dilaria, Luigi Germinario, Claudio Mazzoli, Caterina Previato, Milo K. Pilgrim, Josiah Olah, Jacopo Bonetto, Michele Secco.

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
