## [Editor Report · Decision Letter 0]

8 Jan 2026

Dear Dr. Secco,

Thank you for submitting your manuscript to PLOS ONE. After careful consideration, we feel that it has merit but does not fully meet PLOS ONE’s publication criteria as it currently stands. Therefore, we invite you to submit a revised version of the manuscript that addresses the points raised during the review process.

The manuscript is based on a solid and relevant dataset and addresses an archaeometric problem that falls within the scope of PLOS ONE. However, a number of issues related to the proportionality between analytical resolution and interpretation, the use of absolute priority claims, and the clarity and presentation of figures, tables and Supporting Information need to be addressed before the manuscript can meet the journal’s publication standards. For these reasons, a Major Revision is required, focused on rewording, structural clarification and editorial improvements rather than on the collection of additional data.

We look forward to receiving your revised manuscript.

Kind regards,

Przemysław Mroczek, Dr. hab.

Academic Editor

PLOS One

Journal Requirements:

2. In your manuscript, please provide additional information regarding the specimens used in your study. Ensure that you have reported human remain specimen numbers and complete repository information, including museum name and geographic location.

For more information on PLOS One's requirements for paleontology and archeology research, see https://journals.plos.org/plosone/s/submission-guidelines#loc-paleontology-and-archaeology-research....

MS was supported with the contribution of Fondazione Cassa di Risparmio di Padova e Rovigo as part of the Bando Ricerca Scientifica di Eccellenza 2023 [grant number 68051]. Further support was obtained in the framework of “The Geosciences for Sustainable Development” project (Budget Ministero dell'Università e della Ricerca–Dipartimenti di Eccellenza 2023–2027 C93C23002690001). The research infrastructures employed in this project were implemented and funded by the University of Padova within the World Class Research Infrastructures (WCRI) programme –SYCURI (Synergic Strategies for Cultural Heritage at Risk).

6. We note that Figures 1, 2, and S4 in your submission contain map/satellite images which may be copyrighted. All PLOS content is published under the Creative Commons Attribution License (CC BY 4.0), which means that the manuscript, images, and Supporting Information files will be freely available online, and any third party is permitted to access, download, copy, distribute, and use these materials in any way, even commercially, with proper attribution. For these reasons, we cannot publish previously copyrighted maps or satellite images created using proprietary data, such as Google software (Google Maps, Street View, and Earth). For more information, see our copyright guidelines: http://journals.plos.org/plosone/s/licenses-and-copyright.

a. You may seek permission from the original copyright holder of Figures 1, 2, and S4 to publish the content specifically under the CC BY 4.0 license.

Additional Editor Comments:

A clear, descriptive title, together with a focused abstract and well-chosen keywords, is essential not only for accurately representing the study but also for facilitating the identification of appropriate expert reviewers, which has become an increasingly significant challenge. For this reason, the points raised below place particular emphasis on the title, abstract and keywords, as these elements strongly shape both discoverability and the peer-review process.

While the title broadly reflects the subject of the manuscript, its rhetorical opening (“Beyond the Vitruvian Pozzolans”) is stylistically more interpretative than descriptive and does not adequately reflect the strongly analytical and archaeometric character of the study. As currently formulated, the title places disproportionate emphasis on a conceptual narrative rather than on the material, methods and provenance focus that constitute the core contribution of the paper. In line with PLOS ONE’s preference for clear and descriptive titles, the authors are therefore asked to revise the title towards a more technically explicit, method- and material-oriented formulation. The alternative titles suggested below are provided solely as illustrative examples and should not be considered prescriptive.

Archaeometric provenance of pozzolanic aggregates in Roman mortars from the Euganean Hills (NE Italy)

Identification of a Roman pozzolan quarry in the Euganean Hills Volcanic District (NE Italy): an archaeometric approach

The abstract accurately reflects the content of the manuscript but is overly detailed and method-heavy for a PLOS ONE Research Article. In particular, the level of analytical and interpretative detail exceeds what is typically expected in an abstract and tends to obscure the core research question and main findings. Please streamline the abstract by reducing technical detail, moderating absolute claims, and focusing more clearly on the research aim, principal results and their significance. A revised version of the abstract is provided below for the authors’ consideration:

“This study investigates the provenance of pozzolanic aggregates used in Roman lime-based mortars from the theatre–bath complex at Via Scavi in Montegrotto Terme (ancient Fons Aponi, northeastern Italy), dated to the Early Imperial period. An integrated archaeometric approach combining petrographic observations, mineralogical analyses and bulk geochemical data was applied to characterise the volcanic components of the mortars and assess their hydraulic behaviour. The results show that the mortars incorporate angular trachytic to trachyandesitic volcanic breccias displaying well-developed reaction rims and extensive pozzolanic reactivity, leading to the formation of calcium–aluminosilicate hydrate phases typical of hydraulic lime mortars. Comparison with a comprehensive reference database of volcanic rocks from the Euganean Hills indicates that these aggregates are consistent with explosive diatreme breccias exposed in the eastern sector of the volcanic district, most likely corresponding to quarry areas near Villa Draghi and Via Scagliara. The identification of similar volcanic materials in mortars from Aquileia further suggests that Euganean pozzolans were not used exclusively at the local scale. These findings provide new archaeometric evidence for the exploitation of non-Vitruvian volcanic resources in Roman construction and illustrate the potential of integrated petrographic and geochemical approaches for provenance studies of ancient mortar aggregates.”

The current set of keywords largely repeats terms already present in the title and abstract. While this is not formally incorrect, it limits their usefulness for indexing and discoverability. Please consider revising the keywords so that they complement rather than duplicate the title, for example by emphasising analytical approaches, material properties or technological processes relevant to the study. By way of illustration, more informative keywords could include terms such as archaeometry, hydraulic lime mortars, pozzolanic reaction, volcanic aggregates, petrography, bulk geochemistry, or Roman construction technology.

The following points address several interconnected issues related to the interpretation, proportionality and presentation of the results, with the aim of strengthening the robustness and clarity of the manuscript’s main conclusions.

1/ Priority claims (“first identification”)

The manuscript repeatedly refers to the first identification of a Roman pozzolan quarry in the Euganean Hills. While the study is clearly novel in terms of analytical resolution and dataset size, such absolute priority claims are potentially problematic in a field with a long history of archaeometric research. Please revise the wording throughout the manuscript to avoid categorical “ first” statements and instead emphasise the robustness and resolution of the provenance attribution achieved here.

2/ Use of bulk XRF for heterogeneous breccias

A substantial part of the provenance argument relies on bulk xrf analyses of small, texturally heterogeneous breccia clasts. Although the limitations of this approach are acknowledged, bulk geochemistry is still used to support relatively fine-scale source discrimination. Please clarify more explicitly what level of spatial resolution bulk XRF can realistically support in this context and ensure that all provenance-related conclusions remain strictly proportional to this resolving power.

3/ Interpretation of LDA results

The manuscript correctly notes that Linear Discriminant Analysis forces classification into predefined groups. However, LDA outcomes are later used to argue for the coexistence of material from different quarry systems within single mortars. Please clarify how these probabilistic classifications should be interpreted and avoid drawing conclusions that rely too directly on forced assignments produced by the statistical model.

4/ Degree of quarry-scale discrimination

In several passages, the narrative suggests discrimination at the level of individual quarry sites (Villa Draghi versus Via Scagliara). Given the acknowledged overlap in petrographic and geochemical characteristics, this level of precision may exceed what the data can robustly support. Please revise these sections to make clear where provenance attribution remains probabilistic or ambiguous rather than definitive.

5/ Evidence for medium-distance distribution

The identification of similar pozzolanic material in Aquileia is presented as evidence for medium-distance distribution. While this observation is interesting and potentially significant, it is based on a limited number of samples and contexts.

Please temper the language used here and clarify whether the data support systematic distribution, episodic transport, or isolated movement of materials.

6/ Attribution of technical “expertise” to Roman builders

The manuscript attributes the selection of specific volcanic materials to the empirical expertise of Roman builders and stone masons. While this is a plausible interpretation, it cannot be directly demonstrated by the presented data. Please rephrase such statements more cautiously, making clear where they represent interpretative inferences rather than empirically demonstrable conclusions.

7/ Balance between analytical detail and research focus

Some sections, particularly those describing QPA-XRPD results and multi-parameter geochemical plots, are very detailed and may obscure the main research questions for non-specialist readers. Please consider streamlining these sections and relocating secondary analytical detail to the Supporting Information, ensuring that the main text remains clearly focused on the core objectives of the study.

8/ Repetition of metodological limitations

Limitations related to clast size, alteration processes, bulk geochemical resolution and statistical uncertainty are discussed repeatedly in different sections of the manuscript. Please consolidate these points into a single, clearly structured discussion of methodological constraints to avoid redundancy while maintaining transparency.

The overall structure of the manuscript is appropriate; however, the distinction between the Results and Discussion sections is not always clearly maintained. In several parts of the Results, the presentation of analytical outputs (e.g. bulk geochemical compositions, QPA-XRPD phase proportions and multivariate classification results) is directly followed by interpretative statements concerning provenance attribution, quarry selection or technological choices, which would be more appropriately developed in the Discussion. Conversely, parts of the Discussion reiterate descriptive elements of the Results, including mineral phase abundances, compositional ranges and clustering patterns, rather than focusing on their broader archaeological and technological implications. Please consider sharpening the separation between these sections by keeping the Results strictly focused on empirical observations and analytical outcomes, and confining interpretation, comparison with other studies and contextualisation to the Discussion.

The Conclusions are generally consistent with the results presented; however, their current wording is overly generic and stylistically very smooth, with several broad statements that go beyond the specific empirical scope of the study. Please consider tightening the Conclusions by reducing generalised phrasing, focusing more explicitly on the key findings of this case study, and avoiding formulaic expressions that do not add substantive content.

Figure 1 provides valuable geological and spatial context, but it is currently very information-dense and visually complex. In particular, panel (a) is effectively unreadable at publication scale due to excessively small font sizes and an overabundance of detailed labels, which undermines its function as a general orientation map. This panel would benefit from substantial generalisation, including the reduction of text, simplification of symbology and a clearer visual hierarchy. More broadly, the figure attempts to convey multiple levels of information simultaneously, and the authors are therefore encouraged to simplify the overall design or consider splitting the figure into separate panels focusing on regional context and quarry-scale detail in order to improve clarity and readability.

Figure 2 is accompanied by a caption that is currently too brief and generic to allow the figure to be understood without extensive reference to the main text. The caption does not clearly explain what is shown, how the sampling locations relate to specific architectural elements, or how the site plan connects to the broader regional context. In addition, the figure contains an unexplained black area in the upper left corner, which is not described in the caption or legend and appears to have no clear informational function. The different components of the figure (e.g. site plan, sampling locations, regional inset) should also be clearly identified and labelled with letters (a, b, c) and explicitly referred to in a revised, more informative caption, in line with standard figure presentation practices.

The Supporting Information provides extensive and technically detailed descriptions of analytical instrumentation and procedures; however, its role in relation to the main manuscript is not clearly defined. At present, the S1 File reads as a stand-alone laboratory report rather than as a supplement explicitly designed to support and extend the Methods section. Please clarify the scope and purpose of the Supporting Information at the outset and ensure that it complements, rather than duplicates, the methodological descriptions provided in the main text.

The Supporting Information shows some unevenness in language and formatting that may be noticed during review or production. This includes minor typographical issues and inconsistencies, such as the use of Horneblende instead of Hornblende, the incorrect chemical formula BaSO₃ instead of BaSO₄, and inconsistent notation of units (e.g. mm² versus mm2). In addition, parts of the text retain a very raw laboratory-report style and would benefit from light editorial smoothing. While these issues do not affect the scientific validity of the work, they detract from the overall clarity and presentation of the Supporting Information and should be addressed.

Both Supporting Information spreadsheets contain valuable analytical data, but their current presentation includes several editorial and formatting issues that may be flagged during review or production. In both files, column headings are not always explicit or consistently formatted, with some columns insufficiently labelled, making it difficult to distinguish between oxides, trace elements and calculated parameters. Units and abbreviations are not uniformly defined, and the overall structure of the tables does not always allow the reader to quickly understand the organisation of the datasets. While these issues do not affect the scientific validity of the data, they detract from clarity and reusability and should be addressed by revising the tables to ensure clear headings, consistent units and a more transparent layout suitable for Supporting Information in PLOS ONE.

The manuscript is based on a solid dataset and addresses an interesting and relevant problem; the points raised above are intended to strengthen the clarity, proportionality and presentation of the study rather than to question its overall scientific merit.

---

## [Author Response · Author response to Decision Letter 1]

23 Jan 2026

the responses are reported in the rebuttal letters

---

## [Decision Letter · Decision Letter 1]

2 Mar 2026

Dear Dr. Secco,

Thank you for submitting your manuscript to PLOS ONE. After careful consideration, we feel that it has merit but does not fully meet PLOS ONE’s publication criteria as it currently stands. Therefore, we invite you to submit a revised version of the manuscript that addresses the points raised during the review process.

Both reviewers agree that the study is technically sound, methodologically robust, and supported by appropriate analytical data, and I concur with this assessment. However, several clarifications and corrections are required for acceptance, including consistent volcanological terminology, clarification of specific methodological aspects (XRPD amorphous quantification, LDA validation, effects of acid treatment prior to XRF), and resolution of a number of editorial and formatting artefacts; these changes are mandatory. Minor stylistic improvements and additional explanatory details suggested by the reviewers are recommended to enhance clarity but do not affect the overall validity of the results.

We look forward to receiving your revised manuscript.

Kind regards,

Przemysław Mroczek, Dr. hab.

Academic Editor

PLOS One

**Journal Requirements:**

**Additional Editor Comments:**

Dear Authors,

Thank you for submitting the revised version of your manuscript entitled “Discovery of a Roman Quarry for Pozzolanic Aggregates in the Euganean Hills Volcanic District, Northeast Italy: an archaeometric approach” (PONE-D-25-68252R1).

The manuscript has now been assessed by two expert reviewers. Both consider the study technically sound, methodologically rigorous, and supported by appropriate analytical data. The statistical treatment and the Data Availability statement are regarded as satisfactory. The work represents a careful and well-executed archaeometric investigation with a solid analytical basis.

The remaining comments are largely technical and editorial in nature. Before the manuscript can proceed towards acceptance, I ask that you address the following points.

Please revise the keywords so that they complement rather than repeat the wording of the title. Ensure consistent use of volcanological terminology (e.g. pozzolan vs pozzolana; Gulf of Pozzuoli vs Gulf of Naples; consistent use of “Euganean pozzolans”, with or without quotation marks). Please also ensure that the background discussion of Neapolitan Yellow Tuff and related volcanic units is volcanologically accurate and internally consistent (e.g. NYT linked to the Campi Flegrei system rather than Somma-Vesuvius) and that the use of pulvis puteolanus is correctly associated with the Gulf of Pozzuoli in all relevant passages. Where you use the term “tuff”, please ensure it is applied appropriately (i.e. avoid formulations such as “unconsolidated tuffs” unless clearly justified); if you refer to loose pyroclastic materials, please use a more precise expression and distinguish local “pozzolana” sensu stricto from the broader term “pozzolan(s)”.

Correct minor issues in mineral nomenclature and reporting, including consistent lower-case mineral names throughout the manuscript and Supporting Information, clarification/definition of Afm phases at first use, clarification of the term “lime lumps”, and the use of carbonatic rather than carbonate where appropriate. Please also revise the caption of Supplementary Table S2 to include all analysed mineral phases (amphibole, biotite, plagioclase, magnetite), not only magnetite.

With regard to XRPD, you state that amorphous and crystalline contents were quantified through the addition of 20 wt% zincite as internal standard. For clarity, please explicitly state whether the amorphous fraction was calculated using the internal standard approach rather than by difference. Given the substantial amorphous contents reported in some samples, a brief clarification of the procedure adopted and any steps taken to ensure robustness of the refinements would be helpful.

Concerning XRF sample preparation, mortar clasts were treated with 3% HCl for 24 hours prior to analysis. While you note the possible effect on Y and exclude it from comparative analyses, please clarify whether the potential impact of acid treatment on other mobile elements was evaluated, and briefly state how you ensured that the trace element signature used for provenance comparison was not materially biased by the leaching procedure. Please also correct the typographical error “form XRF data” to “from XRF data”.

You report the use of Linear Discriminant Analysis for provenance discrimination. Please indicate whether any form of model validation (e.g. leave-one-out or cross-validation) was performed and briefly report the outcome. This will help readers assess the robustness of the discriminant model and interpret the reported classification performance appropriately.

There appears to be an inconsistency in the reported Raman laser power (3.0 mW versus 5 mW). Please clarify and ensure consistency throughout the manuscript and Supporting Information. Where magnetite/maghemite discrimination is discussed, please ensure that the evidential basis is presented consistently across methods (Raman and, where relevant, XRPD) and that the role of each technique is clearly stated.

Please also address the remaining detailed points raised by Reviewer 2, including clarification regarding augite identification, the reporting/quantification of smectite and gypsum (including how these phases were quantified if they are listed in tables), correction of “vughs” to “vugs” , and correction of any terminology in Table 1 (including the use of “revetments”, if not appropriate). In addition, please ensure that any duplicated or repetitive text has been removed.

Finally, before resubmission, please carefully proofread the entire manuscript and Supporting Information to remove all editing artefacts and formatting remnants. The current version contains multiple traceable track-change/production artefacts (e.g. duplicated words, merged tokens, repeated “FigureFig” constructions, and residual formatting strings). These issues must be fully resolved in the next version, as they affect readability and presentation quality.

Given the limited scope of the issues outlined above, the manuscript is suitable for publication pending minor revision. Please submit a clean revised version together with a detailed, point-by-point response indicating how each comment has been addressed. I do not anticipate the need for a further round of external review provided that the revisions are implemented clearly and comprehensively.

I look forward to receiving your revised manuscript.

Yours sincerely,

Przemysław Mroczek

Academic Editor

Reviewers' comments:

Reviewer's Responses to Questions

**Comments to the Author**

Reviewer #1: (No Response)

Reviewer #2: (No Response)

2. Is the manuscript technically sound, and do the data support the conclusions?

Reviewer #1: Yes

Reviewer #2: Yes

3. Has the statistical analysis been performed appropriately and rigorously?

Reviewer #1: Yes

Reviewer #2: Yes

4. Have the authors made all data underlying the findings in their manuscript fully available?

Reviewer #1: Yes

Reviewer #2: Yes

5. Is the manuscript presented in an intelligible fashion and written in standard English?

Reviewer #1: Yes

Reviewer #2: Yes

Reviewer #1: The manuscript PONE-D-25-68252_R1, titled “Discovery of a Roman Quarry for Pozzolanic Aggregates in the Euganean Hills Volcanic District, Northeast Italy: an archaeometric approach” by S. Dilaria and co-authors, is a very engaging study that aims to establish the provenance of pozzolanic aggregates used in Roman mortars from the theatre–bath complex at Via Scavi in Montegrotto Terme, dating to the Early Imperial period. Furthermore, the paper offers important data to detect the use of these volcanic materials in mortars from distant locations, such as Aquileia, indicating that “Euganean pozzolans” were not used solely at the local level.

The paper is a revised version of the original submission that appears well-structured, clearly outlining the main findings of the study while maintaining strong scientific rigour and comprehensive results. In my view, the authors’ revisions and responses adequately address the valuable comments raised during the initial review process and, undoubtedly, merit publication in PLOS ONE.

I have only two observations that should not delay the revision process.

I echo the academic editor’s comment, which advised authors to revise the keywords, so they complement rather than duplicate the title. The authors responded that they modified the keywords following the comment, but it seems to me that the keywords remain unchanged: Euganean Hills; Pozzolans; Roman mortars and binders; Provenance analysis; Roman Quarring and Trade.

The caption of the supplementary S2 Table “Results of SEM-EDS analyses on magnetites of breccia clasts from quarry sites of Villa Draghi and Via Scagliara and archaeological samples from Via Scavi in Montegrotto” should include all investigated minerals (Amphibole, Biotite, Plagioclases, Magnetites) names, as it only mentions magnetite.

Sincerely yours,

Alberto De Bonis

Reviewer #2: Abstract: I already had a concern: aggregates sometimes define nor reactive ingredients, so calling them pozzolanic could be misleading.

Line 20: Punctual geochemical data by fluorescence?

Line 21 (microRaman) instead of (mRaman) or use symbol

Line 48: Unconsolidated volcanic tuffs: if using tuffs the term unconsolidated is wrong, it is better to use “non-lithified cineritic deposits”

Line 48: NYT deposits are linked to CF, not to Somma-Vesuvius, so Authors should be consistent with volcanological features

Line 55: Baia quarries are in the Gulf of Pozzuoli not in the Gulf of Naples

Line 59: Authors should be consistent: the term pozzolana indicates the unconsolidated part of the NYT, pozzolan(s) is more general term and indicates all volcanic deposits with pozzolanic activities.

Line 73: … century CE, previously reported as c. BCE

Line 90: once again pulvis puteolanus is from the Gulf of Pozzuoli

Table 1: Pozzolan type column reports seldom chemical compositions of the materials and could be sometimes misleading; please correct the term “revetments”

Line 193: please clarify if lump refers to lime lumps

Line 202: please remove the word stone

Line 210: feldspar or plagioclase?

Line 214: please use carbonatic instead of carbonate

Line 226: please add references for standard mortars

Line 227 vugs not vughs

Line 249: no need to specify the valence of Mg (not previously reported for other cations)

Line 267: please define Afm phases, not reported previously

Line 277-279: please do not use capital letter for minerals

Table 3: The second decimal place (and the first too, IMHO) is completely useless. Are errors available? Amorphous content was estimated by difference? Rietveld analyses of materials with such amount of amorphous is tricky, any further clarification (e.g. PONKS approach, peak phases addition) is available?

Line 361-374: It seems to me to be a repetition of what is stated in lines 200 to 211.

386-388: Authors should comment on presences, not absences.

Line 391: Augite is reported, please clarify its role and above all, its precise identification.

Line 393: Any other info on smectite type?

Table 4: Gypsum is reported, not discussed in the text; please clarify how smectite was quantified

Line 459: please define fresh minerals

Line 485: see comment of line 21

Line 485-491: magnetite/maghemite could be clearly investigated by XRD (FWHM of the main peaks); any evidence through this technique?

Line 560: Please maintain consistency in the text with the term “Euganean pozzolans,” which is sometimes enclosed in quotation marks and sometimes not.

Supporting information: please do not use capital letters for minerals

Line 888: see comment of line 21

.

Reviewer #1: **Yes:** Alberto De BonisAlberto De BonisAlberto De BonisAlberto De Bonis

Reviewer #2: **Yes:** Piergiulio CappellettiPiergiulio CappellettiPiergiulio CappellettiPiergiulio Cappelletti

---

## [Author Response · Author response to Decision Letter 2]

24 Mar 2026

To the Editors of PLoS ONE Journal,

please find attached the reviewed version of our manuscript with the slightly revised title “Discovery of a Roman Quarry for Pozzolanic Aggregates in the Euganean Hills Magmatic District, Northeast Italy: a stepwise archaeometric approach” by myself

and co-authors, for publication in PLoS ONE Journal.

The answers to the comments of both the two reviewers and editor are reported hereafter (in italics).

Academic Editor

• Concerning XRF sample preparation, mortar clasts were treated with 3% HCl for 24 hours prior to analysis. While you note the possible effect on Y and exclude it from comparative analyses, please clarify whether the potential impact of acid treatment on other mobile elements was evaluated, and briefly state how you ensured that the trace element signature used for provenance comparison was not materially biased by the leaching procedure. Please also correct the typographical error “form XRF data” to “from XRF data”.

We thank the Reviewer for this important observation regarding the 3% HCl treatment. Our approach was based on a conservative selection of tracers to guarantee the reliability of the provenance data. Being aware that the exact extent of leaching cannot be quantified for every element, we proactively excluded Yttrium (Y) from all discriminant models, as it appeared most susceptible to mobilization, as observed in similar case studies tailored to the analyses of volcanic aggregates included in mortars (e.g. D’Ambrosio et al. 2015; Marra et al. 2016). In fact, the provenance analysis relied on elements characterized by lower geochemical mobility (e.g., Zr, Nb, Th), which are hosted within stable silicate lattices or resistant accessory minerals. These phases are significantly less affected by mild acid leaching than elements associated with the carbonatic binder. Moreover, the suitability of the selected tracers is empirically validated by the high statistical coherence of the archaeological dataset. In case the leaching procedure would have been materially biased the results, we would have observed high dispersion ("geochemical noise") and a lack of correlation with the sources. Instead, the archaeological clasts maintained sharp clustering and strong affinity with the geological references.

Typographical error “form XRF data” to “from XRF data”: corrected.

• You report the use of Linear Discriminant Analysis for provenance discrimination. Please indicate whether any form of model validation (e.g. leave-one-out or cross-validation) was performed and briefly report the outcome. This will help readers assess the robustness of the discriminant model and interpret the reported classification performance appropriately.

We thank the editor for this pertinent suggestion. To ensure the highest degree of statistical transparency and to assess the robustness of our provenance assignments, we have further validated the Linear Discriminant Analysis (LDA) using Leave-One-Out Cross-Validation (LOOCV). This procedure yielded a 100% correct classification rate for the reference training set (M. Castellone vs. Villa Draghi) in both the "full variable" and "stepwise forward" configurations. This confirms that the geochemical separation between the two source quarries is exceptionally sharp and is not an artifact of overfitting. However, to provide a deeper assessment of the archaeological samples (MG group), we intentionally compared the results of the Full Model (10 variables) with a Stepwise Forward Model. While the reference sources remained stable (100% accuracy), this comparison highlighted the sensitivity of specific archaeological clasts (e.g., MG09_c4 and MG05_c1) to different statistical configurations. Therefore, we have clarified in the manuscript that these divergent outcomes for certain clasts are not merely statistical artifacts but stem from inherent material-specific ambiguities. Factors such as the limited size and heterogeneity of the clasts, the analytical resolution of bulk XRF, and post-depositional pozzolanic alterations can deviate geochemical signatures from their primary values.

We have profoundly revised this part of the manuscript comparing different LDA algorithms. Crucially, the observed variability in provenance attribution between the two models does not stem from statistical instability, but rather from the intrinsic nature and 'quality' of the processed archaeological data. Regardless of the cross-validation outcomes, the resolution of provenance assignments is fundamentally constrained by the geochemical representativeness of the bulk clast profile. At this analytical level, the primary signature of an archaeological sample can be significantly influenced by factors such as its limited size within the mortar matrix and its susceptibility to post-depositional alterations. In essence, the fluctuations in posterior probabilities reflect the 'noise' inherent to the archaeological material itself, where pozzolanic reactions and heterogeneity can deviate the clast from its pristine geological parent, rather than a flaw in the LDA modeling. Consequently, the statistical ambiguity is a transparent reflection of the material's complexity.

• There appears to be an inconsistency in the reported Raman laser power (3.0 mW versus 5 mW). Please clarify and ensure consistency throughout the manuscript and Supporting Information.

Edited and corrected in the supporting file.

• Where magnetite/maghemite discrimination is discussed, please ensure that the evidential basis is presented consistently across methods (Raman and, where relevant, XRPD) and that the role of each technique is clearly stated.

We thank the reviewer for this helpful comment. In the revised manuscript we have clarified the evidential basis used to discuss the discrimination between magnetite and maghemite and the specific role of each analytical technique. XRPD analyses identify magnetite as the dominant Fe-oxide phase in the samples. No maghemite reflections were detected in the diffraction patterns, which is likely due to its very low abundance and/or its occurrence as a minor alteration product below the detection limit of XRPD. Micro-Raman analyses, however, locally revealed spectral features consistent with maghemite, which we interpret as the result of partial oxidation of magnetite. To avoid ambiguity, the revised text now explicitly states that the term “magnetite” is used throughout the manuscript to refer to the main Fe-oxide phase identified by XRPD, while the occurrence of maghemite is discussed only in relation to the Raman data, where it is detected as a secondary alteration product.

• Finally, before resubmission, please carefully proofread the entire manuscript and Supporting Information to remove all editing artefacts and formatting remnants. The current version contains multiple traceable track-change/production artefacts (e.g. duplicated words, merged tokens, repeated “FigureFig” constructions, and residual formatting strings). These issues must be fully resolved in the next version, as they affect readability and presentation quality.

We performed a complete polishing of the English in the text and we revised minor internal inconsistencies in the formatting and referencing of the paper

Reviewer 1

• I echo the academic editor’s comment, which advised authors to revise the keywords, so they complement rather than duplicate the title. The authors responded that they modified the keywords following the comment, but it seems to me that the keywords remain unchanged: Euganean Hills; Pozzolans; Roman mortars and binders; Provenance analysis; Roman Quarring and Trade.

Edited and modified into: hydraulic lime mortars, volcanic aggregates, geochemistry, Roman construction technology, stone provenance, magnetite

• The caption of the supplementary S2 Table “Results of SEM-EDS analyses on magnetites of breccia clasts from quarry sites of Villa Draghi and Via Scagliara and archaeological samples from Via Scavi in Montegrotto” should include all investigated minerals (Amphibole, Biotite, Plagioclases, Magnetites) names, as it only mentions magnetite.

Edited and corrected

Reviewer 2

• Abstract: I already had a concern: aggregates sometimes define nor reactive ingredients, so calling them pozzolanic could be misleading.

Edited and corrected

• Line 20: Punctual geochemical data by fluorescence?

Corrected

• Line 21 (microRaman) instead of (mRaman) or use symbol

Corrected along the entire text as µ-Raman

• Line 48: Unconsolidated volcanic tuffs: if using tuffs the term unconsolidated is wrong, it is better to use “non-lithified cineritic deposits”

Corrected

• Line 48: NYT deposits are linked to CF, not to Somma-Vesuvius, so Authors should be consistent with volcanological features

Corrected

• Line 55: Baia quarries are in the Gulf of Pozzuoli not in the Gulf of Naples

Corrected along the text

• Line 59: Authors should be consistent: the term pozzolana indicates the unconsolidated part of the NYT, pozzolan(s) is more general term and indicates all volcanic deposits with pozzolanic activities.

Corrected into volcanic pozzolans

• Line 73: … century CE, previously reported as c. BCE

This is correct: the first evidence of local use points to the 2nd c BCE while the large diffusion occurs only during the Imperial period

• Line 90: once again pulvis puteolanus is from the Gulf of Pozzuoli

Corrected along the text

• Table 1: Pozzolan type column reports seldom chemical compositions of the materials and could be sometimes misleading; please correct the term “revetments”

Revetement corrected into coating; chemical definition of pozzolan types has been removed.

• Line 193: please clarify if lump refers to lime lumps

Corrected

• Line 202: please remove the word stone

Edited using the definition “lithic materials”

• Line 210: feldspar or plagioclase?

We prefer to maintain the definition of feldspars as they comprise both plagioclases and scattered k-feldspars

• Line 214: please use carbonatic instead of carbonate

Corrected

• Line 226: please add references for standard mortars

Removed the mention to standard mortars

• Line 227 vugs not vughs

Corrected

• Line 249: no need to specify the valence of Mg (not previously reported for other cations)

Corrected

• Line 267: please define Afm phases, not reported previously

The sentence has been edited with a more complete definition of AFm phases

• Line 277-279: please do not use capital letter for minerals

Done

• Table 3: The second decimal place (and the first too, IMHO) is completely useless.

Corrected using only one decimal.

• Are errors available? Amorphous content was estimated by difference? Rietveld analyses of materials with such amount of amorphous is tricky, any further clarification (e.g. PONKS approach, peak phases addition) is available?

The amorphous fraction was calculated using the internal standard approach, following the widely tested procedure described in S1 File.

• Line 361-374: It seems to me to be a repetition of what is stated in lines 200 to 211.

This is the description of the geological samples, while before we were describing the clasts in the archaeological mortars. For the sake of clarity we preferred to provide both the descriptions highlighting the (minimal) differences between archaeological clasts and geological samples.

• 386-388: Authors should comment on presences, not absences.

We added this sentence to highlight the main differences between the clasts and the Euganean volcanic lavas. We think that this sentence is useful to provide mineralogical elements of discrimination of the trachitic breccias in respect to the common lavas.

• Line 391: Augite is reported, please clarify its role and above all, its precise identification.

We removed the mentioning of the augitic type.

• Line 393: Any other info on smectite type?

We refined this phase using a smectite profile from BGMN. We suspect it is a montmorillonite type, but we did not clearly state this.

• Table 4: Gypsum is reported, not discussed in the text; please clarify how smectite was quantified.

Gypsum is interpreted as a secondary phase formed through the sulphation of the exposed bedrock. Smectite was quantified using BGMN, applying refinements specifically tailored to improve the modelling of low-angle clay mineral reflections.

• Line 459: please define fresh minerals

Removed fresh in the description of the minerals

• Line 485: see comment of line 21

done

• Line 485-491: magnetite/maghemite could be clearly investigated by XRD (FWHM of the main peaks); any evidence through this technique?

In the present study, magnetite was identified as the main Fe-oxide phase by Raman, although its reflections are weak and indicate a low abundance in the bulk samples. No clear evidence of maghemite was detected in the XRPD patterns, and therefore the discrimination between magnetite and maghemite based on peak position or FWHM was not possible. This is likely due to the very low concentration of Fe-oxide phases in the samples, which are close to the detection limit of XRPD. Evidence of maghemite was instead obtained only through point micro-Raman analyses, which locally revealed spectral features consistent with maghemite. These occurrences are interpreted as minor oxidation products derived from the alteration of pristine magnetite. We have clarified this point in the revised manuscript, explicitly stating that maghemite was detected only locally by micro-Raman spectroscopy and was not identified by XRPD due to its very low abundance.

• Line 560: Please maintain consistency in the text with the term “Euganean pozzolans,” which is sometimes enclosed in quotation marks and sometimes not.

Done, always adopted the quotation marks

• Supporting information: please do not use capital letters for minerals

Done

• Line 888: see comment of line 21

Done

---

## [Editor Report · Decision Letter 2]

26 Mar 2026

Dear Dr. Secco,

Thank you for submitting your manuscript to PLOS ONE. After careful consideration, we feel that it has merit but does not fully meet PLOS ONE’s publication criteria as it currently stands. Therefore, we invite you to submit a revised version of the manuscript that addresses the points raised during the review process.

The manuscript presents a sound and well-documented multi-analytical approach, and the key technical points raised during peer review have been satisfactorily addressed, including clarification of analytical procedures and validation of the statistical approach. However, further revision is required to meet PLOS ONE’s criteria in terms of clarity and coherence of presentation. In its current form, structural redundancy and an overextended descriptive style reduce readability and weaken the clarity of the main analytical arguments, particularly those concerning provenance. The required revisions therefore focus on improving the organisation and presentation of the existing material rather than introducing new data, so that the conclusions are communicated more clearly and remain fully supported by the evidence.

As the corresponding author, your ORCID iD is verified in the submission system and will appear in the published article. PLOS supports the use of ORCID, and we encourage all coauthors to register for an ORCID iD and use it as well. Please encourage your coauthors to verify their ORCID iD within the submission system before final acceptance, as unverified ORCID iDs will not appear in the published article. *Only* the individual author can complete the verification step; PLOS staff the individual author can complete the verification step; PLOS staff the individual author can complete the verification step; PLOS staff the individual author can complete the verification step; PLOS staff *cannot* verify ORCID iDs on behalf of authors.verify ORCID iDs on behalf of authors.verify ORCID iDs on behalf of authors.verify ORCID iDs on behalf of authors.

We look forward to receiving your revised manuscript.

Kind regards,

Przemysław Mroczek, Dr. hab.

Academic Editor

PLOS One

Journal Requirements:

Additional Editor Comments:

The manuscript requires further tightening at the level of structure and style.

At present, there is a clear overlap between sections, particularly between the Results and the geological/background descriptions.

Petrographic and mineralogical characteristics are in part repeated rather than progressively developed, which weakens the analytical flow of the paper. These sections should be more clearly differentiated, with Results focusing on the material analysed and the geological context used only where it directly supports interpretation. In its current form, this issue affects not only readability but also the clarity of the analytical argument.

In addition, the text is currently somewhat overextended. Descriptive passages are often longer than necessary and, in places, do not add proportionate analytical value. This gives the impression of an “overwritten” manuscript and makes it harder to identify the main findings.

Please revise the text with a view to:

1/ removing or consolidating repeated descriptions across sections (e.g. overlapping petrographic descriptions appearing both in the Results and in the geological background sections),

2/ shortening descriptive passages that do not directly advance the interpretation (for instance, extended mineralogical listings that are not subsequently used in the provenance argument),

3/ sharpening the link between observations and conclusions (i.e. making more explicit how specific petrographic or geochemical features support the provenance assignments).

A more concise and better-structured presentation will substantially improve readability and allow the main results-particularly the provenance arguments-to come through more clearly.

---

## [Author Response · Author response to Decision Letter 3]

28 Mar 2026

The observations raised by the academic editor has been resolved as reported in the rebuttal letter

---

## [Editor Report · Decision Letter 3]

30 Mar 2026

Discovery of a Roman Quarry for Pozzolanic Aggregates in the Euganean Hills Magmatic District, Northeast Italy: a stepwise archaeometric approach

PONE-D-25-68252R3

Dear Dr. Secco,

We’re pleased to inform you that your manuscript has been judged scientifically suitable for publication and will be formally accepted for publication once it meets all outstanding technical requirements.

Kind regards,

Przemysław Mroczek, Dr. hab.

Academic Editor

PLOS One

Additional Editor Comments (optional):

Thank you for submitting the revised version of your manuscript. The paper has improved substantially following revision and now presents a clear and methodologically sound archaeometric study. The integrated multi-analytical approach is well executed, and the provenance interpretation is convincingly supported by the petrographic and geochemical data.

The manuscript meets the criteria for publication in PLOS ONE, in particular with respect to technical soundness and the alignment between data, analyses and conclusions.

No further revisions are required.
---

## [Editor Report · Acceptance letter]

PONE-D-25-68252R3

PLOS One

Dear Dr. Secco,

I'm pleased to inform you that your manuscript has been deemed suitable for publication in PLOS One. Congratulations! Your manuscript is now being handed over to our production team.

Kind regards,

on behalf of

Dr. hab. Przemysław Mroczek

Academic Editor

PLOS One